# REGRET RATES FOR $\epsilon$-GREEDY STRATEGIES FOR NON-PARAMETRIC BANDITS WITH DELAYED REWARDS

## ABSTRACT

Incorporating delayed feedback is often crucial in applying multi-armed bandit algorithms in real-world sequential decision making problems. In this paper, we present finite-time regret upper bounds for $\epsilon$-greedy type allocation strategies in a flexible nonparametric contextual bandits framework with delayed rewards. The strategies presented differ in how the exploration rate changes as a function of delays. We consider unbounded random delays and use the Nadaraya-Watson estimator for estimating the mean reward functions. We also propose practical data-driven strategies that adaptively choose between the two proposed strategies.

## 1 INTRODUCTION

Multi-armed bandit is a sequential decision making problem with the goal of optimally choosing from a set of available arms (or treatments) such that the accumulated sum of rewards received over time is maximized. In this problem, the learner makes a sequence of choices (or actions) from amongst the arms and observes the rewards corresponding to those choices. In addition to this, in most decision making problems, one has access to side information (or covariates) which can aid the decision-making. This framework is then known as contextual bandits or multi-armed bandits with covariates. The first paper on contextual bandits by Woodroofe (1979) was motivated by its application to clinical trials. Contextual bandit algorithms provide a natural framework in any situation where treatment decisions need to be made to optimize some health outcome for the present patients, as has been considered by, Lai et al. (1985); Lai & Liao (2012); Lai et al. (2019); Sklar et al. (2021); Lu et al. (2021). These problems have been studied in both parametric and nonparametric frameworks, see Tewari & Murphy (2017) for a comprehensive review. Most of the bandit algorithms assume instantaneous observance of rewards, but in most practical situations like mobile health and precision medicice, rewards are only obtained at some delayed time. It is often the case that many patients have to be treated before the outcome for the current patient is observed. Below, we review the existing literature on standard and contextual bandits with delayed rewards.

In the standard setting (without covariates), delayed rewards have been studied previously by Dudik et al. (2011); Joulani et al. (2013), where the former consider constant known delay, while the latter provides a systemic study of online learning problems with random delayed feedback. Joulani et al. (2013) developed meta-algorithms which in a black-box fashion could use algorithms developed for the non-delayed case into the ones that can handle delays in a feedback loop. Then, Mandel et al. (2015) devise a method that guarantees good black-box algorithms when leveraging a prior dataset and incorporating heuristics to help improve empirical performance of the algorithms. More recently, Gael et al. (2020) relax the assumptions made in previous works and allow the delay distributions to vary across arms, and consider cases where the delays are heavy-tailed. In the same spirit, Lancewicki et al. (2021) further relax these assumptions on delay distributions using a regular and a phased version of successive elimination approach for the reward independent and dependent case, respectively. More recently, the problem of experts with arm-dependent delays in the non-stochastic case has been studied by Van Der Hoeven & Cesa-Bianchi (2022). Other works on multi-armed bandits with delayed rewards include Cella & Cesa-Bianchi (2020); Guha et al. (2010); Eick (1988). Delayed rewards have also been studied in the adversarial setting (Cesa-Bianchi et al. (2016); Li et al. (2019); Thune et al. (2019); Zimmert & Seldin (2020); Gyorgy & Joulani (2021)) and the delayed anonymous composite feedback setting (Pike-Burke et al. (2017; 2018); Cesa-Bianchi et al. (2018)).

Given that delayed rewards are ubiquitous in a lot of practical applications, there is also growing interest in contextual bandits with delayed rewards. Motivated by delayed conversions in advertising, Vernade et al. (2017; 2020) consider potentially infinite stochastic delays, where the latter deals with the delayed linear bandit problem (contextual) and does not assume prior knowledge of the delay distribution unlike the former. Zhou et al. (2019) designed delay-adaptive algorithm for generalized

linear contextual bandits using UCB-style exploration. Desautels et al. (2014) use Gaussian process bandits and developed algorithms for parallelizing exploration-exploitation trade-offs. More recently, Vakili et al. (2023) have studied UCB strategies for kernel bandits with delayed rewards. Arya & Yang (2020; 2021) consider potentially infinite delays in nonparametric bandits. They provide strong consistency results for the proposed randomized allocation strategies ($\epsilon$-greedy) in the former and present a case for taking into account the extent of delays and problem complexity in delayed contextual bandits in the latter. However, they do not have results for finite-time regret performance and our goal is to study that in this paper. Our focus lies in the study of $\epsilon$-greedy algorithms due to their ease of implementation and potential for good practical performance in various situations, given appropriate exploration probability choices Dann et al. (2022),Bietti et al. (2021). Despite their practical appeal and frequent selection as top choices in real-world scenarios, they have not been extensively studied in the existing literature. Another motivation for investigating $\epsilon$-greedy algorithms is that they employ a randomization scheme, reminiscent of classical randomization approaches used in clinical trials. In addition, our choice to study the non-parametric setting stems from the modeling flexibility it offers, as it allows for non-linear and complicated mean reward functions.

**Contribution:** We study $\epsilon$-greedy type randomized allocation strategies for nonparametric bandits with random unbounded delayed feedback. We present two competing strategies that differ in how the underlying exploration probability sequence is updated and derive finite time regret bounds for them. We obtain sub-linear regret rates depending on the extent of delays. While bounding the estimation error follows a similar path as Qian & Yang (2016) with carefully integrating delays in the analysis, bounding the randomization error in the presence of unbounded delayed rewards is more challenging and is a key theoretical contribution of our work. Another advantage of our work is that it allows stochastic unbounded delays with a relaxed distributional assumption as compared to the existing literature. In our knowledge, this is the first work presenting regret bounds for $\epsilon$-greedy in nonparametric bandits with delayed feedback setting. In addition, from a more practical point of view, we propose two new data-driven schemes that select between the two proposed strategies such that the resulting strategy is advantageous in most situations. We conduct simulation studies to examine the performance of these algorithms under different data generating scenarios.

**Organization:** The rest of the paper is organized as follows. In Section 2, we describe the problem setup of contextual bandits with delayed rewards. In Section 3, we state the two proposed randomized strategies ($\epsilon$-greedy type). Subsequently, in Section 4, we define the Nadaraya-Watson estimator and specify the assumptions made on the model and kernels used in the estimation. Then, the main theorems for finite time regret bounds for the two strategies are in Section 5, followed by a discussion and comparison of the regret rates for the two strategies in Section 6. In Section 7, the adaptive schemes are proposed and we conduct simulation studies to show the improvement in the rate of regret decay by using the adaptive strategies.

## 2 PROBLEM SETUP

Assume that there are $\ell \geq 2$ arms available for allocation. Each arm allocation results in a reward which is obtained at some random time after the arm allocation. Although this setup holds generally, let us describe it from the point of view of treatment allocation. Suppose that for a specific disease, there are $\ell$ competing treatments to be allocated to patients as they visit a doctor. For each patient indexed by $j = 1, 2, \ldots, N$, visiting at known times $s_j \in \mathbb{R}^+$, a treatment $I_j$ is alloted based on previously observed data and the covariate (or context), $X_j$. We assume that the covariates are $d$-dimensional continuous random variables and take values in the hypercube $[0, 1]^d$. Since the rewards can be obtained at some delayed time, we denote $\{t_j \in \mathbb{R}^+, 1 \leq j \leq N\}$ to be the observation time for the rewards for arms $\{I_j, 1 \leq j \leq N\}$ respectively. Let $Y_{i,j}$ denote the reward obtained at time $t_j \geq s_j$ for arm $i = I_j$. Let $f_i(X_j), 1 \leq i \leq \ell$ denote the mean reward for the $i$th arm with covariate $X_j$. The observed reward with covariate $X_j$ by pulling the $i$th arm is modeled as,

$$Y_{i,j} = f_i(X_j) + \epsilon_{ij}, \tag{1}$$

where $\epsilon_{ij}$ denotes random error with $\mathrm{E}(\epsilon_{ij}) = 0$ and $\mathrm{Var}(\epsilon_{ij}) < \infty$ for $j \in \mathbb{N}$ and $i = 1, \ldots, \ell$. The functions $f_i, i = 1, \ldots, \ell$ are unknown and are estimated nonparametrically as described in section 4. Note that our setup is applicable more widely, for example, in settings such as online advertisement recommendations.

Since the rewards are observed at delayed times $\{t_j; 1 \leq j \leq N\}$, the delay in the reward for arm $I_j$ pulled at the $j^{\text{th}}$ time is given by a random variable, $d_j := t_j - s_j$. We assume that these delays are independent of both the covariates and arms. That is, let $d_j \sim G_j, j \geq 1$ be independent random

variables with $G_j$ the probability distribution for $j^{\text{th}}$ delay. Let $\tau_n = \sum_{j=1}^{n} I(t_j \leq n)$ denote the number of rewards observed by time $n$. Note that $\tau_n$ is a random variable and would be used often in our algorithms and results.

Let $\{X_j, j \geq 1\}$ be a sequence of covariates independently generated according to an unknown underlying probability distribution $P_X$, from a population supported in $[0,1]^d$. We denote $\eta$ to be a sequential allocation strategy, which for each time $j$ chooses an arm $I_j$ based on the previous observations and $X_j$. The total mean reward up to time $n$ is $\sum_{j=1}^{n} f_{I_j}(X_j)$. To evaluate the performance of the allocation strategy, let $i^*(x) = \arg\max_{1 \leq i \leq \ell} f_i(x)$ and $f^*(x) = f_{i^*(x)}(x)$. Without the knowledge of the random errors, the ideal performance occurs when the choices of arms selected $I_1, \ldots, I_n$ match the optimal arms $i^*(X_1), \ldots, i^*(X_n)$, yielding the optimal total reward $\sum_{j=1}^{n} f^*(X_j)$. Thus we measure the performance of the allocation strategy, $\eta$, by the regret, $R_n(\eta) = \sum_{j=1}^{n} f^*(X_j) - f_{I_j}(X_j)$. Note that, we obtain a sub-linear regret rate if $\frac{R_n(\eta)}{n} \to 0$ as $n \to \infty$ with probability 1, and finite time analysis provides an upper bound on the rate of this decay.

## 3 THE PROPOSED STRATEGIES

In this section, we present the proposed allocation strategies for which we will derive the regret upper bounds. Define $Z^{n,i}$ to be the set of observations for arm $i$ whose rewards have been obtained up to time $n-1$, that is, $Z^{n,i} := \{(X_j, Y_{i,j}) : 1 \leq t_j \leq n-1 \text{ and } I_j = i\}$. Let $\hat{f}_{i,n}$ denote the regression estimator of $f_i$ using a regression method based on the data $Z^{n,i}$. Let $\{\pi_j, j \geq 1\}$ be a sequence of positive numbers in $[0,1]$ decreasing to zero, such that $(\ell-1)\pi_j < 1$ for all $j \geq 1$. We propose two strategies $\eta_1$ and $\eta_2$ with a subtle difference in the arm selection step but same algorithmic structure. In the algorithms above, step 1 initializes the allocations by pulling each arm alternatively until

---

**Algorithm 1** Randomized allocation with delayed rewards

1: Allocate arms randomly until we have at least one reward observed for each arm. Suppose, that happens at time $m_0$.
2: **for** $n = m_0 + 1, \ldots, N$ **do**
3:       *Estimate the individual functions $f_i$.* For $n = m_0 + 1$, based on $Z^{n,i}$, estimate $f_i$ by $\hat{f}_{i,n}$ for $1 \leq i \leq \ell$ using the chosen regression procedure.
4:       *Best-performing arm (projected).* For $X_n$, let $\hat{i}_n(X_n) = \arg\max_{1 \leq i \leq \ell} \hat{f}_{i,n}(X_n)$.
5:       *Select and pull.* The arm pulled is given by:

     a) Strategy $\eta_1$: $I_n = \begin{cases} \hat{i}_n, & \text{with probability } 1 - (\ell-1)\pi_n \\ i, & \text{with probability } \pi_n, \ i \neq \hat{i}_n, \ 1 \leq i \leq \ell. \end{cases}$

     b) Strategy $\eta_2$: $I_n = \begin{cases} \hat{i}_n, & \text{with probability } 1 - (\ell-1)\pi_{\tau_n} \\ i, & \text{with probability } \pi_{\tau_n}, \ i \neq \hat{i}_n, \ 1 \leq i \leq \ell. \end{cases}$

6:      *Update the estimates.*

     a) If one or more rewards are obtained at the $n^{\text{th}}$ time, update the function estimates of $f_i$ for the respective arms.

     b) If no reward is obtained at the $n^{\text{th}}$ time, use $\hat{f}_{i,n+1} = \hat{f}_{i,n} \ \forall \ i \in \{1, \ldots, \ell\}$.

7: **end for**

---

we observe at least one reward for each arm. Step 3 estimates the mean reward function for each arm. This could be done using several regression methods, and we use Nadaraya-Watson regression estimator as described in Section 4. Steps 4 and 5 enforce an $\epsilon$-greedy type of randomization scheme which prefers the projected best performing arm so far with some probability and explores with the remaining. The preference is determined by a user determined sequence of exploration probability $\{\pi_n, n \geq 1\}$, which for strategy $\eta_2$ only gets updated when a new reward is observed, that is, $\pi_{\tau_n}$. While for strategy $\eta_1$, it is updated at every time point irrespective of a reward being observed or not, that is, $\pi_n$. Hence, the two strategies differ in the extent of exploration and exploitation that is allowed over time. Finally in step 6, the mean reward function estimators are updated if new rewards are observed or they remain the same if no new rewards are observed.

## 4 REGRESSION ESTIMATOR

We focus on Nadaraya-Watson regression for estimating the mean reward functions, $f_i, 1 \leq i \leq \ell$, in both the proposed allocation strategies $\eta_1$ and $\eta_2$. We choose $h_{\tau_n}$ for the bandwidth sequence, where

the subscript of $\tau_n$ (running index of the number of rewards observed by time $n$) means that we only update the bandwidth when a new reward is observed. This choice is logical as it would make sense to reduce the bandwidth only when new data is observed.

For arm $1 \le i \le \ell$, at each time point $n$, define $J_{i,n} = \{1 \le j \le n-1 : I_j = i, 1 \le t_j \le n-1\}$, be the indices corresponding to the rewards that were observed for that arm by time $n-1$. Let $\mathcal{A}_N = \{(s_j, t_j) : t_j \le N, 1 \le j \le N\}$, denote the pair of time points at which arms were allotted (known) and at which corresponding rewards were obtained (random) by time $N$, respectively. Note that, given $\mathcal{A}_N$, we would exactly know the delay in observing a reward at each allocation. Also, let $X^n = \sigma\langle X_1, X_2, \ldots, X_n \rangle$ denote the sigma-field generated by the covariates until time $n$.

Recall that, the Nadaraya-Watson estimator of $f_i(x)$ is,

$$\hat{f}_{i,n+1}(x) = \frac{\sum_{j \in J_{i,n+1}} Y_{i,j} K\left(\frac{x-X_j}{h_{\tau_n}}\right)}{\sum_{j \in J_{i,n+1}} K\left(\frac{x-X_j}{h_{\tau_n}}\right)}. \tag{2}$$

Given $x \in [0,1]^d$, $1 \le i \le \ell$ and $n \ge m_0 + 1$, define $Q_{n+1}(x) = \{1 \le j \le n : 1 \le t_j \le n, \|x-X_j\|_\infty \le Lh_{\tau_n}\}$ and $Q_{i,n+1}(x) = \{1 \le j \le n : 1 \le t_j \le n, I_j = i, \|x-X_j\|_\infty \le Lh_{\tau_n}\}$. In other words, these are the indices for the observed rewards in the a local bin containing $X_j$ and corresponding to arm $i$ respectively. We use these sets in the proofs for Theorems 1 and 2. Let $M_{n+1}(x)$ and $M_{i,n+1}(x)$ be the size of $Q_{n+1}(x)$ and $Q_{i,n+1}(x)$, respectively.
If for a given time instance $n$ and arm $i$, the denominator of the Nadaraya-Watson estimator in equation 2 is extremely small, we will replace the kernel $K(\cdot)$ in equation 2 with a uniform kernel $I(\|u\|_\infty \le L)$. In particular for the case when the complement of the event $B_{i,n}$ defined as,

$$B_{i,n}^c := \left\{ \frac{1}{M_{i,n+1}(x)} \sum_{j \in J_{i,n+1}} K\left(\frac{x-X_j}{h_{\tau_n}}\right) < c_5 \right\} \tag{3}$$

occurs almost surely for some small positive constant $0 < c_5 < 1$, we will use the uniform kernel. Next, we present the assumptions required to establish the regret upper bound.

### 4.1 ASSUMPTIONS

We start by making some assumptions on the errors, the underlying functions, the kernel function used in the definition of Nadaraya-Watson estimator in equation 2 and the delays.

**Assumption 1.** *The errors satisfy a (conditional) moment condition that there exists $v, c > 0$ and $c$ such that for all integers $k \ge 2$, $i \in \{1, \ldots, \ell\}$ and $n \ge 1$, $\mathbb{E}(|\epsilon_{in}|^k | X_n) \le \frac{k!}{2} v^2 c^{k-2}$, almost surely.*

This assumption imposes some moment conditions on the error distributions known as the refined Bernstein condition (as in Birgé et al. (1998); Qian & Yang (2016)). Assumption 1 is met for a wide range of distributions, for example, normal distribution and bounded errors, making it viable in a wide range of applications. In the Supplementary files, we also consider sub-Exponential errors and establish the corresponding regret upper bounds. Next, we consider two natural assumptions on the mean reward functions and the covariate density, respectively. Although we restrict the covariate space to $[0,1]^d$, any bounded and compact subset of $\mathbb{R}^d$ would suffice.

**Assumption 2.** *The functions $f_i$ are continuous on $[0,1]^d$ with, $A := \sup_{1 \le i \le \ell} \sup_{x \in [0,1]^d} (f^*(x) - f_i(x)) < \infty$.*

**Assumption 3.** *The design distribution $P_X$ is dominated by the Lebesgue measure with a continuous density $p(x)$ uniformly bounded above and away from 0 on $[0,1]^d$; that is, $p(x)$ satisfies $\underline{c} \le p(x) \le \bar{c}$ for $0 < \underline{c} \le \bar{c}$.*

In other words, Assumption 3 guarantees that the contexts are sampled with a positive probability across the entire domain of $[0,1]^d$. Next, for Kernel regression, we consider a multivariate nonnegative kernel function $K(u) : \mathbb{R}^d \to \mathbb{R}$ that satisfies Lipschitz, boundedness and bounded support conditions. Note that these are standard assumptions made in nonparametric regression literature (see Theorem 1.8, Tsybakov (2004)).

**Assumption 4.** *For some constants $0 < \lambda < \infty$, $|K(u) - K(u')| \le L\|u - u'\|_\infty$, for all $u, u' \in \mathbb{R}^d$.*

**Assumption 5.** *There exists constants $L_1 \le L$, $c_3 > 0$ and $c_4 \ge 1$ such that $K(u) = 0$ for $\|u\|_\infty > L$, $K(u) \ge c_3$ for $\|u\|_\infty \le L_1$ and $K(u) \le c_4$ for all $u \in \mathbb{R}^d$.*

Next, we make assumptions on the delays. Assumption 6 is an independence assumption on the delays which is sensible in many applications. Assumption 7 mildly restricts the expected number of delayed rewards such that we expect to observe an increasing number of rewards as time progresses.

**Assumption 6.** *The delays, $\{d_j, j \geq 1\}$, are independent of each other, the arms and the covariates.*

**Assumption 7.** *The partial sums of delay distributions satisfy, $\sum_{j=1}^{n} G_j(n - s_j) = \Omega(q(n))$, where $q(n)$ is a sequence such that $q(n) \to \infty$ as $n \to \infty$.*

This assumption is not restrictive as it allows for rewards to be unbounded as long as a minimum number of rewards are being observed in finite time. More precisely, based on condition equation 8 and equation 10, the result holds as long as $q(n)$ grows faster than $\log n$, we can choose $h_n$ and $\pi_n$ to be such that the conditions equation 8 and equation 10 hold, respectively. In essence, this implies that we can achieve sub-linear regret rates for both the proposed strategies, provided that the expected number of observed rewards by time $n$ grows strictly faster than $\log n$ for sufficiently large values of $n$. This assumption would naturally hold for a lot of scenarios with delayed rewards where some informed learning is plausible.

## 5 FINITE-TIME RESULTS

In this section we present finite time upper bounds for the cumulative regret for both strategies $\eta_1$ and $\eta_2$. The proofs for all the results stated in this section can be found in the Supplementary files. To characterize the underlying function class being considered for the mean reward functions, we define the modulus of continuity, $w(h; f_i)$.

**Definition 1.** *Modulus of continuity: For some $h > 0$,*

$$w(h; f_i) = \sup\{|f_i(x_1) - f_i(x_2)| : ||x_1 - x_2||_\infty \leq h\}. \tag{4}$$

**Lemma 1.** *Under Assumption 6 and Assumption 7, $\tau_n \overset{a.s.}{\to} \infty$ as $n \to \infty$.*

**Lemma 2** (For strategy $\eta_2$). *Suppose Assumptions 1,2, 5 and 6 are satisfied and $\{\pi_n\}$ is a decreasing sequence. Given $x \in [0, 1]^d$, $1 \leq i \leq \ell$ and $n \geq m_0 + 1$, for every $\epsilon > w(Lh_{\tau_n}; f_i)$ a.s., we have for strategy $\eta_2$,*

$$P_{X^n, \mathcal{A}_N}^{\eta_2}(|\hat{f}_{i,n+1}(x) - f_i(x)| \geq \epsilon) \leq \exp\left(-\frac{3M_{n+1}(x)\pi_{\tau_n}}{28}\right)$$
$$+ 4N \exp\left(-\frac{c_5^2 M_{n+1}(x)\pi_{\tau_n}(\epsilon - w(Lh_{\tau_n}; f_i))^2}{4c_4^2 v^2 + 4c_4 c(\epsilon - w(Lh_{\tau_n}; f_i))}\right) \tag{5}$$

*where $P_{X^n, \mathcal{A}_N}^{\eta_2}(\cdot)$ denotes the conditional probability for strategy $\eta_2$ given the design points $X^n = \sigma\langle X_1, \ldots, X_n\rangle$, $\mathcal{A}_N = \{(s_j, t_j); t_j \leq N, 1 \leq j \leq N\}$ and $\tau_n = \sum_{j=1}^{n} I\{t_j \leq n\}$, which is a known quantity given $\mathcal{A}_N$.*

**Lemma 3** (For strategy $\eta_1$). *Suppose Assumptions 1,2, 5 and 6 are satisfied and $\{\pi_n\}$ is a decreasing sequence. Given $x \in [0, 1]^d$, $1 \leq i \leq \ell$ and $n \geq m_0 + 1$, for every $\epsilon > w(Lh_{\tau_n}; f_i)$ a.s., we have for strategy $\eta_1$,*

$$P_{X^n, \mathcal{A}_N}^{\eta_1}(|\hat{f}_{i,n+1}(x) - f_i(x)| \geq \epsilon) \leq \exp\left(-\frac{3M_{n+1}(x)\pi_n}{28}\right)$$
$$+ 4N \exp\left(-\frac{c_5^2 M_{n+1}(x)\pi_n(\epsilon - w(Lh_{\tau_n}; f_i))^2}{4c_4^2 v^2 + 4c_4 c(\epsilon - w(Lh_{\tau_n}; f_i))}\right), \tag{6}$$

*where $P_{X^n, \mathcal{A}_N}^{\eta_1}(\cdot)$ denotes the conditional probability for strategy $\eta_1$ given the design points $X^n = \sigma\langle X_1, \ldots, X_n\rangle$ and $\mathcal{A}_N = \{(s_j, t_j); t_j \leq N, j \geq 1\}$ and $\tau_n = \sum_{j=1}^{n} I\{t_j \leq n\}$, which is a known quantity given $\mathcal{A}_N$.*

It can be seen that Lemma 2 and Lemma 3 only differ in the hyper-parameter choice of $\pi_{\tau_n}$ and $\pi_n$, other things remain the same. The reason for this is that both are conditional probability results, and given $\mathcal{A}_N$, $\tau_n$ is a known quantity. Next, we provide the theorems for finite-time regret bounds on the cumulative regret for strategy $\eta_2$ and $\eta_1$ respectively.

Given $0 < \delta < 1$ and the total time horizon $N$, for strategy $\eta_2$, let,

$$n_\delta' = \min\left\{n > m_0 : \exp\left(-\frac{3\underline{c}\tilde{a}_1(2Lh_{q(n)})^d \pi_{q(n)} q(n)}{112}\right) \leq \frac{\delta}{4\ell N}\right\}. \tag{7}$$

**Theorem 1.** *Suppose Assumptions 1-7 are satisfied and $\{\pi_n\}$ is a decreasing sequence. Assume $N > n'_\delta$ from equation 7, for the kernel estimator in equation 2 and equation 3. Choose, $\{h_n\}_n$ and $\{\pi_n\}_n$ such that,*

$$\frac{h_{q(n)}^{2d}\pi_{q(n)}^4 q(n)}{\log n} \to \infty, \ as \ n \to \infty. \tag{8}$$

*Then for $0 < \delta \leq 1/4$, we have that, with probability at least $1 - \frac{32\delta}{9}$, the regret for $\eta_2$ satisfies,*

$$R_N(\eta_2) < An'_\delta + \sum_{n=n'_\delta+1}^{N} 2\left(\max_{1 \leq i \leq \ell} w(Lh_{q(n)}; f_i) + \frac{C_{N,\delta}}{\sqrt{h_{q(n)}^d \pi_{q(n)} q(n)}}\right)$$

$$+ A\sum_{t=1}^{N^*(\delta)} M_\delta(\ell-1)\pi_t + \max\left\{A\sqrt{M_\delta \frac{E(\tau_N)}{2}\log\left(\frac{2}{\delta}\right)}, A\sqrt{\left(\frac{N}{2}\right)\log\left(\frac{2}{\delta}\right)}\right\},$$

*where $N^*(\delta) = \mathbb{E}(\tau_N) + \sqrt{\frac{N}{2}\log\left(\frac{1}{\delta}\right)}$, $C_{N,\delta} = \sqrt{64c_4^2 v^2 \log(12\ell N^2/\delta)/c_5^2 \underline{c}(2L)^d}$ and $M_\delta$ is a number chosen such that $\left(1 - \frac{a_1 q(M_\delta/2)}{M_\delta/2}\right)^{M_\delta/2} = \delta$, where $q(.)$ comes from Assumption 7.*

Under the condition equation 8, we have, $n'_\delta/N \to 0$ as $N \to \infty$. Therefore, the regret incurred during the initialization phase is going to be dominated by the regret incurred during the algorithmic phase in the long run. For strategy $\eta_2$, we only update the exploration probability sequence when we observe a new reward. Since delay in observing rewards is a random variable, the maximum distance between consecutive observed rewards plays an important role in bounding the randomization error, as can be seen from the upper bound in Theorem 1.

Now, given $0 < \delta < 1$, for strategy $\eta_1$ and some positive constant $\tilde{\tilde{a}}_1$, let,

$$n''_\delta = \min\left\{n \geq m_0 : \exp\left(-\frac{3\underline{c}\tilde{\tilde{a}}_1(2Lh_{q(n)})^d\pi_n q(n)}{112}\right) \leq \frac{\delta}{4\ell N}\right\}. \tag{9}$$

**Theorem 2.** *Suppose assumptions 1-7 are satisfied and $\{\pi_n\}$ is a decreasing sequence. Assume $N > n''_\delta$ as defined in equation 9 and the kernel estimator as defined in equation 2 and kernel chosen as described in equation 3. We choose, $\{\pi_n\}$ and $\{h_n\}$ so that,*

$$\frac{h_{q(n)}^{2d}\pi_n^4 q(n)}{\log n} \to \infty, \ as \ n \to \infty. \tag{10}$$

*Let $C_{N,\delta} = \sqrt{64c_4^2 v^2 \log(12\ell N^2/\delta)/c_5^2 \underline{c}(2L)^d}$, then with probability larger than $1 - 2\delta$, the cumulative regret for strategy $\eta_1$ satisfies,*

$$R_N(\eta_1) < An''_\delta + \sum_{n=n''_\delta+1}^{N} 2\left(\max_{1 \leq i \leq \ell} w(Lh_{q(n)}; f_i) + \frac{C_{N,\delta}}{\sqrt{h_{q(n)}^d \pi_n q(n)}} + A(\ell-1)\pi_n\right)$$

$$+ A\sqrt{\left(\frac{N}{2}\log\left(\frac{1}{\delta}\right)\right)},$$

Under the condition equation 10, we will have, $n''_\delta/N \to 0$ as $N \to \infty$. Therefore, for large enough time horizon $N$, we will have $N > n''_\delta$.

Also note, when we have no delays, we obtain the same regret rate as in Qian & Yang (2016) for both the strategies $\eta_1$ and $\eta_2$. The right hand side of the inequalities in Theorems 1 and 2 above consists of several terms that are insightful. The first term $An'_\delta$ and $An''_\delta$ comes from the initial rough exploration, respectively. The second term, $\max_{1 \leq i \leq \ell} w(Lh_{q(n)}; f_i)$ is associated with the estimation bias. The third terms in both the results, i.e., $C_{N,\delta}/\sqrt{h_{q(n)}^d \pi_{q(n)} q(n)}$ and $C_{N,\delta}/\sqrt{h_{q(n)}^d \pi_n q(n)}$ can be associated with the estimation standard error, which depends on delay. That is, if the delays are expected to be large, then $q(n)$ will be small as a result of which the estimation standard error will be large. The next term $\sum_{t=1}^{N^*(\delta)} M_\delta(\ell-1)\pi_t$ and $(\ell-1)\pi_n$ is the randomization error, respectively, where $M_\delta$ is a probabilistic upper bound on the difference between consecutive reward

observations. While the former may potentially be quite large for large delay situations leading to large randomization error, the latter is not affected by the delay because as per the proposed allocation strategy, allocations are made at each time point. Finally, the last term in both results is reflective of the fluctuation of the randomization scheme, where the former depends on the extent of delays while the latter does not.

## 6 COMPARISON AND DISCUSSION

As both the upper bounds in Theorem 1 and Theorem 2 consist of components that reflect the bias-variance trade-off and the exploration-exploitation trade-off, we can compare the bounds to get some idea of the underlying nature of the two strategies, $\eta_2$ and $\eta_1$, respectively. In order to compare more specifically, we make an assumption on the class of functions and a specific delay scenario.

**Assumption 8.** *There exist positive constants $\rho$ and $\kappa \leq 1$ such that for each reward function $f_i$, the modulus of continuity satisfies, $\omega(h; f_i) \leq \rho h^\kappa$.*

**Assumption 9.** *Let $E(\tau_N) = O(\sqrt{N})$, i.e., on average we expect to observe about $\sqrt{N}$ many rewards by time $N$.*

Then, we would have $q(N) \leq B\sqrt{N}$ for some constant $B > 0$. Under assumptions 8 and 9, and if we choose $\{\pi_n\} = \frac{1}{\ell-1} n^{-1/(3+d/\kappa)}$ and $\{h_n\} = \frac{1}{L} n^{-1/(3\kappa+d)}$, then we get the following rates.

**Corollary 1.** *Suppose Assumptions 1-9 hold. Then if we choose, $\{\pi_n\} = \frac{1}{\ell-1} n^{-1/(3+d/\kappa)}$ and $\{h_n\} = \frac{1}{L} n^{-1/(3\kappa+d)}$, and for $0 < \delta \leq 1/4$, we have that, with probability at least $1 - \frac{32\delta}{9}$, the cumulative regret for $\eta_2$ satisfies,*

$$R_N(\eta_2) < An_\delta' + 2\left(2\rho + C_{N,\delta}^*\right) N^{\left(1 - \frac{1}{2(3+d/\kappa)}\right)} + AM_\delta^* N^{\frac{1}{2}\left(1 - \frac{1}{(3+d/\kappa)}\right)}$$

$$+ \max\left\{ A\sqrt{M_\delta \frac{\sqrt{N}}{2} \log\left(\frac{2}{\delta}\right)}, A\sqrt{\left(\frac{N}{2}\right) \log\left(\frac{2}{\delta}\right)} \right\}, \tag{11}$$

*where $C_{N,\delta}^* = \sqrt{64c_4^2 v^2 \log(12\ell N^2/\delta)/c_5^2 \underline{c} 2^d}$, $M_\delta^* = M_\delta \left(1 + \sqrt{\log\left(\frac{1}{\delta}\right)}\right)^{1 - \frac{1}{2(3+d/\kappa)}}$.*

*Remark 1:* We can get a bound in expectation using the fact that $\mathbb{E}(R_N(\eta_2)) \leq \int_0^\infty P(R_N(\eta_2 > \zeta)d\zeta$. For instance, consider the scenario where $M_\delta = O(\sqrt{N})$. Under Assumptions 8 and 9, the first term dominates in equation 11, leading to the following result: $\mathbb{E}(R_N(\eta_2)) = O\left(\log(N)N^{\left(1 - \frac{1}{2(3+d/\kappa)}\right)}\right)$. Note that the rate is sub-linear in $N$. This sub-linearity still holds even when the maximum difference between consecutive reward observation times, $M_\delta$, is large.

**Corollary 2.** *Suppose the same Assumptions 1-9 hold. Then if we choose, $\{\pi_n\} = \frac{1}{\ell-1} n^{-1/(3+d/\kappa)}$ and $\{h_n\} = \frac{1}{L} n^{-1/(3\kappa+d)}$, assume $N > n_\delta''$. For $C_{N,\delta}^* = \sqrt{64c_4^2 v^2 \log(12\ell N^2/\delta)/c_5^2 \underline{c} 2^d}$, with probability larger than $1 - 2\delta$, the cumulative regret for strategy $\eta_1$ satisfies,*

$$R_N(\eta_1) < An_\delta'' + 2\left(2\rho N^{\left(1 - \frac{1}{2(3+d/\kappa)}\right)} + C_{N,\delta}^* N^{\left(1 - \frac{1}{4(3+d/\kappa)}\right)} + AN^{\left(1 - \frac{1}{3+d/\kappa}\right)}\right)$$

$$+ A\sqrt{\left(\frac{N}{2} \log\left(\frac{1}{\delta}\right)\right)}. \tag{12}$$

*Remark 2:* Similar to Remark 1, we note that, under Assumptions 8 and 9, the expected regret satisfies $\mathbb{E}(R_N(\eta_1)) = O\left(\log(N)N^{1 - \frac{1}{4(3+d/\kappa)}}\right)$. Importantly, this rate remains independent of $M_\delta$, meaning that regardless of the difference between consecutive reward observation times, we obtain the same rate as long as, on average, we observe a total of $O(\sqrt{N})$ rewards by time $N$.

Note that there is a trade-off in the bounds of the two strategies in equation 11 and equation 12. While the upper bound for the estimation bias (second term) in the two strategies remains the same, the bound on the estimation standard error component (third term) for the former ($\eta_2$) is smaller than the latter ($\eta_1$). However, the randomization error bound (fourth term) for strategy $\eta_2$ is large as compared to the randomization error bound for strategy $\eta_1$ depending on the value of $M_\delta$, which could potentially be of the order $O(N - \sqrt{N})$ in the worst case. If $M_\delta$ is not too large (less than

or equal $O(\sqrt{N})$), we see that the last term corresponding to the fluctuation of the randomization scheme in both the bounds could actually be about the same ($\approx A\sqrt{(N/2)\log{(1/\delta)}}$).

Thus, we notice that the extent to which estimation error or randomization error overpowers the other is also determined by the severity of delays. Note that the rates obtained in theorems 1 and 2, are sub-linear, and fast when $d$ is small and $\kappa$ is close to 1. For instance, when $d = 1, \kappa = 1$, we have the estimation error bound to be of the order, $\tilde{O}(N^{7/8})$ and $\tilde{O}(N^{15/16})$ for $\eta_2$ and $\eta_1$, respectively, when only $O(\sqrt{N})$ rewards are observed by time $N$. Note that it is relevant and important to study randomized allocation strategies because of their easy applicability and good empirical performance. Also, randomized strategies like $\epsilon$-greedy open the doors to answering pertinent questions on statistical inference and robustness for such online-learning algorithms, for example, Chen et al. (2021).

From the finite-time results of Theorem 1 and 2, we note that both strategies $\eta_1$ and $\eta_2$ can be advantageous in different scenarios. This forms the motivation behind development of strategies that can combine the two strategies $\eta_1$ and $\eta_2$ in a data-driven way. As these strategies make decisions locally, we want to take into account the variability in the observed rewards for various arms in a neighborhood of the current covariate, in order to decide between strategy $\eta_1$ and $\eta_2$. In the following section, we propose two adaptive strategies that combine $\eta_1$ and $\eta_2$ in a data-driven fashion. Then, we conduct a simulation study in Section 7 comparing $\eta_1$, $\eta_2$ and the adaptive strategies, $\eta_{\text{adap1}}$ and $\eta_{\text{adap2}}$. We notice that in most situations it is beneficial to use the adaptive strategies as they perform better (or at par) than both $\eta_1$ and $\eta_2$ in reducing the overall regret.

## 7 ADAPTIVE STRATEGIES AND SIMULATION STUDIES

Recall, given $x \in [0,1]^d$, $1 \le i \le \ell$ and $j \ge m_0+1$, $Q_j(x) = \{1 \le k \le j-1 : 1 \le t_k \le j-1, ||x - X_k||_\infty \le Lh_{\tau_j}\}$ and $Q_{i,j}(x) = \{1 \le k \le j-1 : 1 \le t_k \le j-1, I_k = i, ||x - X_k||_\infty \le Lh_{\tau_j}\}$, with their respective sizes given by $M_j(x)$ and $M_{i,j}(x)$. Recall, $\hat{i}_j$ is the arm with the highest estimated mean reward corresponding to covariate $X_j$ at time $j$, and $\tau_n = \sum_{j=1}^n I(t_j \le n)$ is the number of rewards observed by time $n$. Then for the first adaptive strategy, $\eta_{\text{adap}_1}$, we look at the number of observed rewards locally, based on which we determine whether to choose $\eta_1$ or $\eta_2$. For the second strategy, $\eta_{\text{adap}_2}$, instead of using the number of observed rewards locally, we compare the local sample variance of rewards observed in the neighborhood of the current covariate of interest. Let, $\hat{\sigma}^2_{X_j,i} = \text{Sample Variance}\{Y_{i,k} : k \in Q_{i,j}(X_j)\}$ be the sample variance of rewards observed in the bin of side-width $h_{\tau_j}$ around $X_j$. After Step 5 of Algorithm 1, we implement the following to get the new strategies.

**Strategy $\eta_{\text{adap}_1}$:** For $\lambda_1 > 0$, if $\{M_{\hat{i}_j,j}(X_j) > \lambda_1 M_{i,j}(X_j) \text{ for all } i \ne \hat{i}_j, j \le N\}$, then use strategy $\eta_1$, otherwise use strategy $\eta_2$.
**Strategy $\eta_{\text{adap}_2}$:** For $\lambda_2 > 0$, if $\{\hat{\sigma}^2_{X_j,\hat{i}_j} \le \lambda_2 \hat{\sigma}^2_{X_j,i} \text{ for all } i \ne \hat{i}_j, j \le N\}$, then use strategy $\eta_1$, otherwise use $\eta_2$.

For $\eta_{\text{adap}_1}$, the strategy $\eta_1$ is preferred over $\eta_2$ if the number of observations corresponding to a projected best performing arm for that covariate is higher than other arms in a small neighborhood of that covariate. For $\eta_{\text{adap}_2}$, the choice is made when the variance of a projected best performing arm is lower than other arms in a small neighborhood of that covariate. This allows us to avoid unnecessary exploration when we are more confident in our estimates locally. Note that the hyper-parameters, $\lambda_1, \lambda_2 > 0$, are user-determined parameters which are chosen depending on the problem.

**Simulation study:** We conduct a simulation study to compare the per-round average regret for strategies $\eta_1, \eta_2, \eta_{\text{adap}_1}$ and $\eta_{\text{adap}_2}$ under different delayed rewards scenarios. We assume $d = 2, \ell = 2$, $x \in [0,1]^2$, and the simulations run until time $N = 8000$ with first 30 rounds of initialization. For each of the setups, we define one-dimensional functions $g_1$ and $g_2$, and then for $x_1, x_2 \in [0,1]$, we define, $f_1(x_1, x_2) = g_1(x_1) * x_2$ and $f_2(x_1, x_2) = g_2(x_1) * x_2$. The one dimensional functions $g_i$ for each of these setups are plotted in the leftmost panel of Figure 1.
**Setup 1:** In this setup, we consider two well-separated sinusoidal functions, where one is a shifted above version of the other. $g_1(x) = (-2\sin(20\pi x) + 3)$, $g_2(x) = (-2\sin(20\pi x) + 2)$; $x \in [0,1]$.
**Setup 2:** Consider two sinusoidal functions such that the best arm alternates rapidly as the functions oscillate. $g_1(x) = 2\cos(5\pi x) + 2$, $g_2(x) = -2\sin(5\pi x) + 2$, for $x \in [0,1]$.

We consider the following delay scenarios with delay 2 being more severe than delay 1.
**Delay 1:** Each case has probability 0.7 to delay and the delay is half-normal with scale, $\sigma = 1500$.
**Delay 2:** In this case we increase the number of non-observed rewards. Divide the data into four

equal consecutive parts (quarters), such that, in part 1, we only observe every $10^{\text{th}}$ (with Geom(0.3) delay) observation by time $N$ and not observe the remaining; in part 2, we only observe every $15^{\text{th}}$ reward; in part 3, only observe every $20^{\text{th}}$ reward; in part 4, only observe every $25^{\text{th}}$ reward.

We simulate the data from the above mentioned true mean reward functions in equation 1 where $\epsilon_j \overset{\text{i.i.d.}}{\sim} N(0, \sigma = 0.5)$. We use Nadaraya-Watson estimator with Gaussian kernel in equation 2. We run all four strategies $\eta_1$, $\eta_2$, $\eta_{\text{adap}_1}$ and $\eta_{\text{adap}_2}$ for 60 independent replications for time horizon $N = 8000$. Then the average regret $R_n(\eta)/n$ for each time point also averaged over the replications is plotted in figure 1. Therefore, the faster this goes to zero, the better it is. We consider hyper-parameter sequences, $\{h_n\} = (\log n)^{-1}$ and $\{\pi_n\} = (\log n)^{-1}$, however results from other combinations show similar trends and are included in the Supplementary files.

Note that we can tune the parameter $\lambda_1$ and $\lambda_2$ for both the strategies $\eta_{\text{adap}_1}$ (purple) and $\eta_{\text{adap}_2}$ (pink dashed), respectively, but that is not the focus of this study. Further discussion on this can be found in the supplementary material. We use $\lambda_1 = 1$ for strategy $\eta_{\text{adap}_1}$ for both simulation setups, whereas for strategy $\eta_{\text{adap}_2}$, we use $\lambda_2 = 1$ for setup 2 and $\lambda_2 = 3$ for setup 1 in figure 1. In general for these choices of $\lambda$'s, we notice that the two adaptive strategies performs either better or at par with both strategies $\eta_1$ and $\eta_2$ for both delay scenario 1 and 2.

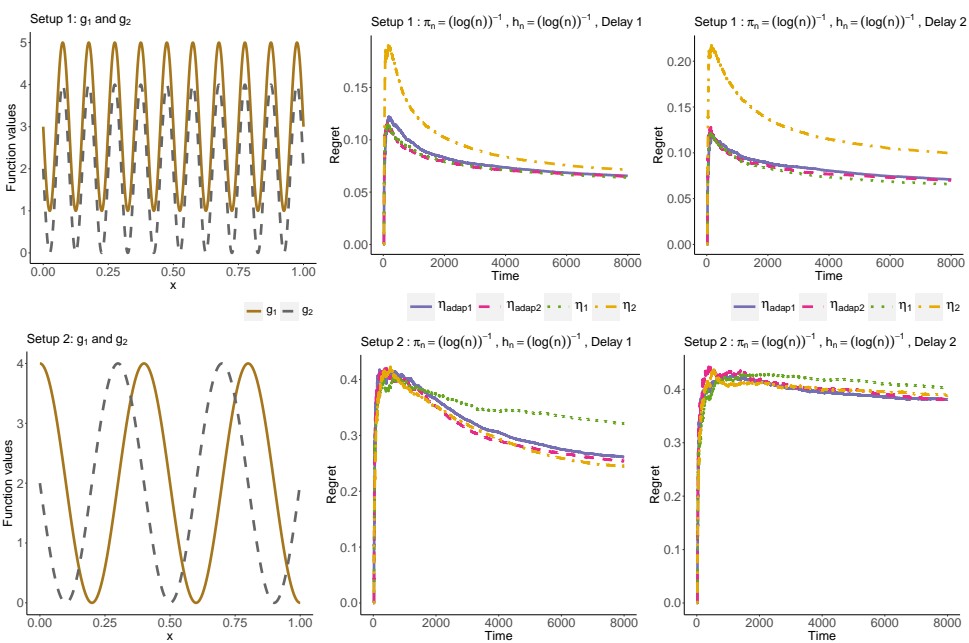

Figure 1: Strategies $\eta_{\text{adap}_1}$ and $\eta_{\text{adap}_2}$ perform better (or at par with) than $\eta_1$ and $\eta_2$.

## 8 CONCLUSION

In this work, we present a finite-time regret analysis for randomized allocation strategies for nonparametric bandits with delayed rewards. Delays are assumed to be independent and unbounded as long as we expect to see a minimum number of observations in finite time. We study finite time regret behavior of the two strategies that essentially differ in how the exploration probability sequence $\{\pi_n\}$ is updated. Based on the finite time upper bounds, we notice that strategy $\eta_2$ leads to lower estimation standard error but higher randomization error, as compared to strategy $\eta_1$. The extent to which one of these competing error term would dominate the other may depend on the severity of delays. Since both the strategies seem to be advantageous in different settings, we propose two adaptive strategies that choose between $\eta_1$ and $\eta_2$ in a data-driven way, based on local behavior of rewards for the arms. However, introducing the adaptive step in these algorithm induces additional dependence structure posing new theoretical challenges which are left for future work. In many practical situations, it may likely be the case that delays depend on the choice of arms and/or the covariates (or context) in the problem. However, new tools and techniques would be required to tackle these problems and would be an interesting future direction to consider.

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
