# Supplementary material: Regret Rates for Randomized Allocation Strategies for Nonparametric Bandits with Delayed Rewards

## S1 Results with sub-exponential errors

In this section, we extend the finite-time regret rates for the two proposed strategies in Section 3 to the case with sub-exponential errors.

**Assumption S1.** *We assume that $\epsilon_j \sim$ Sub-Exponential$(\nu^2, \alpha)$, i.e.,*

$$E(\exp(\lambda \epsilon_j)) \leq \exp\left\{\frac{\lambda^2 \nu^2}{2}\right\}, \ \forall \, \lambda : |\lambda| < \frac{1}{\alpha}.$$

**Theorem S1.** *Suppose assumption S1 and assumptions 2-6 in the paper are satisfied and $\{\pi_n\}$ is a decreasing sequence. Assume that $N > n_\delta^{(3)}$ and the kernel estimator as defined in (2) and kernel chosen as described in (3). Then with probability larger than $1 - 2\delta$, the cumulative regret for strategy $\eta_2$ satisfies,*

$$R_N(\eta_2) < \begin{cases} An'_\delta + \sum_{n=n'_\delta+1}^{N} 2\left(\max_{1 \leq i \leq \ell} w(Lh_{q(n)}; f_i) + \dfrac{C'_{N,\delta}}{\sqrt{h_{q(n)}^d \pi_{q(n)} q(n)}}\right) \\ \quad + A\sum_{t=1}^{N^*(\delta)} M_\delta(\ell-1)\pi_t + \max\left\{A\sqrt{M_\delta \frac{E(\tau_N)}{2}\log\left(\frac{2}{\delta}\right)}, A\sqrt{\left(\frac{N}{2}\right)\log\left(\frac{2}{\delta}\right)}\right\}, \\ \qquad\qquad if \ h_{q(n)}^d q(n)\pi_{q(n)} > \dfrac{8\log(16\ell N^2/\delta)\alpha^2}{\nu^2 c_5 \underline{c}\tilde{a}_1 (2L)^d}, \\[2ex] An'_\delta + \sum_{n=n'_\delta+1}^{N} 2\left(\max_{1 \leq i \leq \ell} w(Lh_{q(n)}; f_i) + \dfrac{C^{ime\prime}_{N,\delta}}{h_{q(n)}^d \pi_{q(n)} q(n)}\right) \\ \quad + A\sum_{t=1}^{N^*(\delta)} M_\delta(\ell-1)\pi_t + \max\left\{A\sqrt{M_\delta \frac{E(\tau_N)}{2}\log\left(\frac{2}{\delta}\right)}, A\sqrt{\left(\frac{N}{2}\right)\log\left(\frac{2}{\delta}\right)}\right\}, \\ \qquad\qquad if \ h_{q(n)}^d q(n)\pi_{q(n)} < \dfrac{8\log(16\ell N^2/\delta)\alpha^2}{\nu^2 c_5 \underline{c}\tilde{a}_1 (2L)^d}, \end{cases}$$

(S1.1)

*where,* $C'_{N,\delta} = \sqrt{8C^2\nu^2 \log(16\ell N^2/\delta)/c_5 \underline{c}\tilde{a}_1(2L)^d}$ *and* $C''_{N,\delta} = 8\alpha C \log(16\ell N^2/\delta)/(c_5 \underline{c}\tilde{a}_1(2L)^d)$.

Note that the modification of sub-exponential errors does not effect the randomization error, however the estimation error changes depending upon the amount of delay and choice of the hyperparameters $\{h_n\}$ and $\{\pi_n\}$. Note that, we will get a similar result for strategy $\eta_1$. The proof can be found in section S3.

 # S2 Proof of Lemmas

 *Proof of Lemma 1.* Recall, $\tau_n = \sum_{j=1}^{n} I\{t_j \leq n\}$. Then, $\mathrm{E}(\tau_n) = \mathrm{E}(\sum_{j=1}^{n} I\{t_j \leq n\}) = \sum_{j=1}^{n} P(t_j \leq n) = \sum_{j=1}^{n} G_j(n - s_j)$. Now, by Assumption 7 we have, for large enough $n$, there exists a positive integer $a_1$, such that, $\sum_{j=1}^{n} G_j(n - s_j) \geq a_1 q(n)$. Then using the inequality in Corollary S2, we get,

$$P\left(\tau_n \leq \frac{a_1 q(n)}{2}\right) \leq P\left(\tau_n \leq \frac{\sum_{j=1}^{n} G_j(n - s_j)}{2}\right)$$

$$\leq \exp\left(-\frac{3\sum_{j=1}^{n} G_j(n - s_j)}{28}\right)$$

$$\leq \exp\left(\frac{-3a_1 q(n)}{28}\right).$$

It is easy to see that the upper bound is summable in $n$ under the condition (10) and (8). By the Borel-Cantelli lemma, the event $\{\tau_n > a_1 q(n)/2\}$ happens infinitely often, therefore $\tau_n \overset{\text{a.s.}}{\to} \infty$. Note that, by construction this implies that $h_{\tau_n} \overset{\text{a.s.}}{\to} 0$, and $\pi_{\tau_n} \overset{\text{a.s.}}{\to} 0$ as $n \to \infty$. As an immediate consequence of this along with continuity of $f$, we get that $w(h_{\tau_n}; f) \overset{\text{a.s.}}{\to} 0$, as $n \to \infty$. $\qquad\square$

## S2.1 Proof of Lemma 2

*Proof of Lemma 2.* Recall that $Q_{n+1}(x) = \{1 \leq j \leq n : 1 \leq t_j \leq n, ||x - X_j|| \leq Lh_{\tau_n}\}$ and $Q_{i,n+1}(x) = \{1 \leq j \leq n : j \in Q_{n+1}(x), I_j = i\}$. Let $M_{n+1}(x)$ and $M_{i,n+1}(x)$ be the size of $Q_{n+1}(x)$ and $Q_{i,n+1}(x)$, respectively. It can be seen that if $M_{n+1}(x) = 0$, (5) trivially holds. Therefore, without loss of generality we can assume $M_{n+1}(x) > 0$. For the event $B_{i,n} = \left\{\frac{1}{M_{i,n+1}(x)} \sum_{j \in J_{i,n+1}} K\left(\frac{x - X_j}{h_{\tau_n}}\right) \geq c_5\right\}$. Note that,

$$P_{X^n, \mathcal{A}_N}\left(|\hat{f}_{i,n+1}(x) - f_i(x)| \geq \epsilon\right)$$

$$= P_{X^n, \mathcal{A}_N}\left(|\hat{f}_{i,n+1}(x) - f_i(x)| \geq \epsilon, \frac{M_{i,n+1}(x)}{M_{n+1}(x)} \leq \frac{\pi_{\tau_n}}{2}\right)$$

$$+ P_{X^n, \mathcal{A}_N}\left(|\hat{f}_{i,n+1}(x) - f_i(x)| \geq \epsilon, \frac{M_{i,n+1}(x)}{M_{n+1}(x)} > \frac{\pi_{\tau_n}}{2}\right)$$

$$\leq P_{X^n, \mathcal{A}_N}\left(\frac{M_{i,n+1}(x)}{M_{n+1}(x)} \leq \frac{\pi_{\tau_n}}{2}\right) + P_{X^n, \mathcal{A}_N}\left(|\hat{f}_{i,n+1}(x) - f_i(x)| \geq \epsilon, \frac{M_{i,n+1}(x)}{M_{n+1}(x)} > \frac{\pi_{\tau_n}}{2}\right)$$

$$\overset{a}{\leq} \exp\left(-\frac{3M_{n+1}(x)\pi_{\tau_n}}{28}\right) + P_{X^n, \mathcal{A}_N}\left(|\hat{f}_{i,n+1}(x) - f_i(x)| \geq \epsilon, \frac{M_{i,n+1}(x)}{M_{n+1}(x)} > \frac{\pi_{\tau_n}}{2}, B_{i,n}\right)$$

$$+ P_{X^n, \mathcal{A}_N}\left(|\hat{f}_{i,n+1}(x) - f_i(x)| \geq \epsilon, \frac{M_{i,n+1}(x)}{M_{n+1}(x)} > \frac{\pi_{\tau_n}}{2}, B_{i,n}^c\right)$$

$$=: \exp\left(-\frac{3M_{n+1}(x)\pi_{\tau_n}}{28}\right) + A_1 + A_2, \tag{S2.1}$$

where the first term in the inequality in step $a$ comes from the extended Bernstein inequality (S6.2).
By Assumption 5 and the definition 1 of the modulus of continuity, we have,

$$
\begin{aligned}
&|\hat{f}_{i,n+1}(x) - f_i(x)| \\
&= \left| \frac{\sum_{j \in J_{i,n+1}} Y_{i,j} K\left(\frac{x-X_j}{h_{\tau_n}}\right)}{\sum_{j \in J_{i,n+1}} K\left(\frac{x-X_j}{h_{\tau_n}}\right)} - f_i(x) \right| \\
&= \left| \frac{\sum_{j \in J_{i,n+1}} (f_i(X_j) + \epsilon_j) K\left(\frac{x-X_j}{h_{\tau_n}}\right)}{\sum_{j \in J_{i,n+1}} K\left(\frac{x-X_j}{h_{\tau_n}}\right)} - f_i(x) \right| \\
&= \left| \frac{\sum_{j \in J_{i,n+1}} (f_i(X_j) - f_i(x)) K\left(\frac{x-X_j}{h_{\tau_n}}\right)}{\sum_{j \in J_{i,n+1}} K\left(\frac{x-X_j}{h_{\tau_n}}\right)} + \frac{\sum_{j \in J_{i,n+1}} \epsilon_j K\left(\frac{x-X_j}{h_{\tau_n}}\right)}{\sum_{j \in J_{i,n+1}} K\left(\frac{x-X_j}{h_{\tau_n}}\right)} \right| \\
&\leq \sup_{x,y: ||x-y||_\infty \leq Lh_{\tau_n}} |f_i(x) - f_i(y)| + \left| \frac{\sum_{j \in J_{i,n+1}} \epsilon_j K\left(\frac{x-X_j}{h_{\tau_n}}\right)}{\sum_{j \in J_{i,n+1}} K\left(\frac{x-X_j}{h_{\tau_n}}\right)} \right| \\
&= w(Lh_{\tau_n}; f_i) + \left| \frac{\sum_{j \in J_{i,n+1}} \epsilon_j K\left(\frac{x-X_j}{h_{\tau_n}}\right)}{\sum_{j \in J_{i,n+1}} K\left(\frac{x-X_j}{h_{\tau_n}}\right)} \right|.
\end{aligned}
\tag{S2.2}
$$

Under $B_{i,n}$,

$$
|\hat{f}_{i,n+1}(x) - f_i(x)| \leq w(Lh_{\tau_n}; f_i) + \frac{1}{c_5 M_{i,n+1}(x)} \left| \sum_{j \in Q_{i,n+1}(x)} \epsilon_j K\left(\frac{x-X_j}{h_{\tau_n}}\right) \right|.
$$

Using this, we will construct an upper bound for $A_1$. Define $\sigma_t = \inf\{\tilde{n} : \sum_{j=1}^{\tilde{n}} I\{I_j = i, t_j \leq n, ||x - X_j|| \leq Lh_{\tau_n}\} \geq t\}, t \geq 1$. Then, for large enough $n$, by Lemma 1, $\epsilon > w(Lh_{\tau_n}, f_i)$ a.s. and we have,

$$
\begin{aligned}
A_1 &\leq P_{X^n, \mathcal{A}_N} \left( \left| \sum_{j \in Q_{i,n+1}(x)} \epsilon_j K\left(\frac{x-X_j}{h_{\tau_n}}\right) \right| \geq c_5 M_{i,n+1}(x)(\epsilon - w(Lh_{\tau_n}; f_i)), \right. \\
&\qquad \left. \frac{M_{i,n+1}(x)}{M_{n+1}(x)} > \frac{\pi_{\tau_n}}{2} \right) \\
&\leq \sum_{\bar{n}=0}^{n} P_{X^n, \mathcal{A}_N} \left( \left| \sum_{t=1}^{\bar{n}} \epsilon_{\sigma_t} K\left(\frac{x-X_{\sigma_t}}{h_{\tau_n}}\right) \right| \geq c_5 \bar{n}(\epsilon - w(Lh_{\tau_n}, f_i)), \right. \\
&\qquad \left. \frac{M_{i,n+1}(x)}{M_{n+1}(x)} > \frac{\pi_{\tau_n}}{2}, M_{i,n+1}(x) = \bar{n} \right) \\
&\leq \sum_{\lceil M_{n+1}(x)\pi_{\tau_n}/2 \rceil}^{n} P_{X^n, \mathcal{A}_N} \left( \left| \sum_{t=1}^{\bar{n}} \epsilon_{\sigma_t} K\left(\frac{x-X_{\sigma_t}}{h_{\tau_n}}\right) \right| \geq c_5 \bar{n}(\epsilon - w(Lh_{\tau_n}; f_i)) \right) \\
&\leq \sum_{\lceil M_{n+1}(x)\pi_{\tau_n}/2 \rceil}^{n} 2 \exp\left( -\frac{\bar{n} c_5^2 (\epsilon - w(Lh_{\tau_n}; f_i))^2}{2c_4^2 v^2 + 2c_4 c(\epsilon - w(Lh_{\tau_n}; f_i))} \right) \\
&\leq 2N \exp\left( -\frac{c_5^2 M_{n+1}(x)\pi_{\tau_n}(\epsilon - w(Lh_{\tau_n}; f_i))^2}{4c_4^2 v^2 + 4c_4 c(\epsilon - w(Lh_{\tau_n}; f_i))} \right),
\end{aligned}
\tag{S2.3}
$$

where the last inequality follows from Lemma S8 and the upper boundedness of the kernel function (assumption 5). Now, to find the bound for $A_2$, under $B_{i,n}^c$ we run into technical problems since the denominator of the Nadaraya-Watson estimator can be extremely small, hence we will replace the

kernel $K(\cdot)$ in (2) with a uniform kernel $I(||u||_\infty \leq L)$. That is for the case when,

$$B_{i,n}^c := \left\{ \sum_{j \in J_{i,n+1}} K\left(\frac{x - X_j}{h_{\tau_n}}\right) < c_5 \sum_{j \in J_{i,n+1}} I(||x - X_j||_\infty \leq Lh_{\tau_n}) \right\}, \qquad \text{(S2.4)}$$

for some small positive constant $0 < c_5 < 1$, we will use the uniform kernel. Therefore, using (S2.2), (S2.4) and (S6.5) (Lemma S8), we get that,

$$A_2 \leq P_{X^n, \mathcal{A}_N}\left(\left|\sum_{j \in J_{i,n+1}} \epsilon_j I(||x - X_j|| \leq Lh_{\tau_n})\right| \geq M_{i,n+1}(x)(\epsilon - w(Lh_{\tau_n}; f_i)),\right.$$

$$\left.\frac{M_{i,n+1}(x)}{M_{n+1}(x)} > \frac{\pi_{\tau_n}}{2}\right)$$

$$\leq \sum_{\bar{n}=0}^{n} P_{X^n, \mathcal{A}_N}\left(\left|\sum_{t=1}^{\bar{n}} \epsilon_{\sigma_t} I(||x - X_{\sigma_t}|| \leq Lh_{\tau_n})\right| \geq \bar{n}(\epsilon - w(Lh_{\tau_n}; f_i)),\right.$$

$$\left.\frac{M_{i,n+1}(x)}{M_{n+1}(x)} > \frac{\pi_{\tau_n}}{2}, M_{i,n+1}(x) = \bar{n}\right)$$

$$\leq \sum_{\lceil M_{n+1}(x)\pi_{\tau_n}/2\rceil}^{n} P_{X^n, \mathcal{A}_N}\left(\left|\sum_{t=1}^{\bar{n}} \epsilon_{\sigma_t} I(||x - X_{\sigma_t}|| \leq Lh_{\tau_n})\right| \geq \bar{n}(\epsilon - w(Lh_{\tau_n}; f_i))\right)$$

$$\leq \sum_{\lceil M_{n+1}(x)\pi_{\tau_n}/2\rceil}^{n} 2\exp\left(-\frac{\bar{n}(\epsilon - w(Lh_{\tau_n}; f_i))^2}{2v^2 + 2c(\epsilon - w(Lh_{\tau_n}; f_i))}\right)$$

$$\leq 2N\exp\left(-\frac{M_{n+1}(x)\pi_{\tau_n}(\epsilon - w(Lh_{\tau_n}; f_i))^2}{4v^2 + 4c(\epsilon - w(Lh_{\tau_n}; f_i))}\right). \qquad \text{(S2.5)}$$

Therefore, using the fact that $0 < c_5 \leq 1 \leq c_4$, (S2.3) and (S2.5) in (S2.1), we get,

$$P_{X^n, \mathcal{A}_N}\left(|\hat{f}_{i,n+1}(x) - f_i(x)| \geq \epsilon\right)$$

$$\leq \exp\left(-\frac{3M_{n+1}(x)\pi_{\tau_n}}{28}\right) + 4N\exp\left(-\frac{c_5^2 M_{n+1}(x)\pi_{\tau_n}(\epsilon - w(Lh_{\tau_n}; f_i))^2}{4c_4^2 v^2 + 4c_4 c(\epsilon - w(Lh_{\tau_n}; f_i))}\right). \qquad \text{(S2.6)}$$

$\square$

The proof for Lemma 3 will follow the same steps with $\pi_{\tau_n}$ replaced by $\pi_n$. Next, we prove a lemma that would be used to prove Theorem 1.

**Lemma S1.** *4 An $\epsilon$ that satisfies,*

$$4N\exp\left(-\frac{c_5^2 \underline{c}\tilde{a}_1(2Lh_{q(n)})^d \pi_{q(n)} q(n)(\epsilon - w(Lh_{q(n)}; f_i))^2}{16c_4^2 v^2 + 16c_4 c(\epsilon - w(Lh_{q(n)}; f_i))}\right) \leq \frac{\delta}{4\ell N}, \qquad \text{(S2.7)}$$

*is given by,*

$$\tilde{\epsilon}_{i,n} = w(Lh_{q(n)}; f_i) + \sqrt{\frac{64c_4^2 v^2 \log(16\ell N^2/\delta)}{c_5^2 \underline{c}\tilde{a}_1(2L)^d h_{q(n)}^d \pi_{q(n)} q(n)}}.$$

*Proof for Lemma S1.* Let $Z := \epsilon - w(Lh_{q(n)}; f_i)$, then (S2.7) becomes,

$$\frac{c_5^2 \underline{c}\tilde{a}_1(2Lh_{q(n)})^d \pi_{q(n)} q(n) Z^2}{16c_4^2 v^2 + 16c_4 c Z} \geq \log\left(\frac{16\ell N^2}{\delta}\right).$$

Let $A_1 = c_5^2 \underline{c}\tilde{a}_1(2L)^d$, $A_2 = 16c_4^2 v^2$, $A_3 = 16c_4 c$.

$$A_1 q(n) h_{q(n)}^d \pi_{q(n)} Z^2 - A_3 \log\left(\frac{16\ell N^2}{\delta}\right) Z - A_2 \log\left(\frac{16\ell N^2}{\delta}\right) \geq 0. \qquad \text{(S2.8)}$$

51 Left hand side is a quadratic polynomial in $Z$. Solving for $Z$,

$$A_1 q(n) h_{q(n)}^d \pi_{q(n)} Z^2 - A_3 \log\left(\frac{16\ell N^2}{\delta}\right) Z - A_2 \log\left(\frac{16\ell N^2}{\delta}\right) = 0$$

$$\Rightarrow Z = \frac{1}{2}\left(\frac{A_3 \log(16\ell N^2/\delta)}{A_1 q(n) h_{q(n)}^d \pi_{q(n)}} \pm \sqrt{\frac{A_3^2 \log^2(16\ell N^2/\delta)}{(A_1 q(n) h_{q(n)}^d \pi_{q(n)})^2} + \frac{4A_2 \log(16\ell N^2/\delta)}{A_1 q(n) h_{q(n)}^d \pi_{q(n)}}}\right).$$

52

53 This will give two real roots for the quadratic equation. Therefore if we want some value of $Z$ such
54 that (S2.8) holds, we can use a point that is larger than the roots $-b \pm \sqrt{b^2 + d^2}$ and we know that
55 $d \geq -b \pm \sqrt{b^2 + d^2}$. Therefore, a potential candidate could be,

$$Z = \sqrt{\frac{4A_2 \log(16\ell N^2/\delta)}{A_1 q(n) h_{q(n)}^d \pi_{q(n)}}}$$

$$= \sqrt{\frac{64 c_4^2 v^2 \log(16\ell N^2/\delta)}{c_5^2 \underline{c} \tilde{a}_1 (2L)^d h_{q(n)}^d \pi_{q(n)} q(n)}},$$

56 which means that we want

$$\tilde{\epsilon}_{i,n} = w(Lh_{q(n)}; f_i) + \sqrt{\frac{64 c_4^2 v^2 \log(16\ell N^2/\delta)}{c_5^2 \underline{c} \tilde{a}_1 (2L)^d h_{q(n)}^d \pi_{q(n)} q(n)}}.$$

57 $\qquad\qquad\qquad\qquad\qquad\qquad\qquad\qquad\qquad\qquad\qquad\qquad\qquad\qquad\qquad\qquad\qquad\qquad\qquad$ □

58 A similar lemma with $\pi_{q(n)}$ replaced by $\pi_n$ could be derived that will be used in the proof of Theorem
59 2.

60 **Lemma S2.** *5 An $\epsilon$ that satisfies,*

$$4N \exp\left(-\frac{c_5^2 \underline{c} \tilde{\tilde{a}}_1 (2Lh_{q(n)})^d \pi_n q(n)(\epsilon - w(Lh_{q(n)}; f_i))^2}{16 c_4^2 v^2 + 16 c_4 c(\epsilon - w(Lh_{q(n)}; f_i))}\right) \leq \frac{\delta}{4\ell N}, \qquad (S2.9)$$

61 *is given by,*

$$\tilde{\epsilon}'_{i,n} = w(Lh_{q(n)}; f_i) + \sqrt{\frac{64 c_4^2 v^2 \log(16\ell N^2/\delta)}{c_5^2 \underline{c} \tilde{\tilde{a}}_1 (2L)^d h_{q(n)}^d \pi_n q(n)}}.$$

62 # S3 Proof of Theorems

63 *Proof of Theorem 1.* By definition of $\hat{i}_j$, $\hat{f}_{i^*(X_j),j} \leq \hat{f}_{\hat{i}_j,j}(X_j)$, then the regret accumulated after the
64 initial forced sampling period is,

$$\sum_{j=m_0+1}^{N} (f^*(X_j) - f_{I_j}(X_j))$$

$$= \sum_{j=m_0+1}^{N} (f_{i^*(X_j)}(X_j) - \hat{f}_{i^*(X_j),j}(X_j) + \hat{f}_{i^*(X_j),j}(X_j) - f_{\hat{i}_j}(X_j)$$

$$+ f_{\hat{i}_j}(X_j) - f_{I_j}(X_j))$$

$$\leq \sum_{j=m_0+1}^{N} (f_{i^*(X_j)}(X_j) - \hat{f}_{i^*(X_j),j}(X_j) + \hat{f}_{\hat{i}_j,j}(X_j) - f_{\hat{i}_j}(X_j)$$

$$+ f_{\hat{i}_j}(X_j) - f_{I_j}(X_j))$$

$$\leq \sum_{j=m_0+1}^{N} (2 \sup_{1 \leq i \leq l} |\hat{f}_{i,j}(X_j) - f_i(X_j)| + AI\{I_j \neq \hat{i}_j\}) \qquad (S3.1)$$

Here the first term corresponds to the regret incurred due to estimation error and the second term corresponds to the randomization error.

We will first find an upper bound for the estimation error. Note that Lemma 2 gives a probability inequality for the estimation error conditional on $\mathcal{A}_N$ and $X^n$. Therefore, in order to get a probability (not conditional) bound on the estimation error, we first remove this condition on $X^n$ and then remove the condition on $\mathcal{A}_N$ in (5). Given arm $i$, for a large enough $n$ satisfying $n \geq m_0 + 1$ and $\epsilon > w(Lh_{\tau_n}; f_i)$ a.s., consider,

$$
P_{\mathcal{A}_N}(|\hat{f}_{i,n+1}(X_{n+1}) - f_i(X_{n+1})| \geq \epsilon)
$$

$$
= P_{\mathcal{A}_N}\left(|\hat{f}_{i,n+1}(X_{n+1}) - f_i(X_{n+1})| \geq \epsilon, M_{n+1}(X_{n+1}) \leq \frac{\underline{c}(2Lh_{\tau_n})^d \tau_n}{2}\right)
$$

$$
+ P_{\mathcal{A}_N}\left(|\hat{f}_{i,n+1}(X_{n+1}) - f_i(X_{n+1})| \geq \epsilon, M_{n+1}(X_{n+1}) > \frac{\underline{c}(2Lh_{\tau_n})^d \tau_n}{2}\right) \quad \text{(S3.2)}
$$

$$
\leq P_{\mathcal{A}_N}\left(M_{n+1}(X_{n+1}) \leq \frac{\underline{c}(2Lh_{\tau_n})^d \tau_n}{2}\right)
$$

$$
+ P_{\mathcal{A}_N}\left(|\hat{f}_{i,n+1}(X_{n+1}) - f_i(X_{n+1})| \geq \epsilon, M_{n+1}(X_{n+1}) > \frac{\underline{c}(2Lh_{\tau_n})^d \tau_n}{2}\right)
$$

$$
\leq \exp\left(-\frac{3\underline{c}(2Lh_{\tau_n})^d \tau_n}{28}\right) + \exp\left(-\frac{3\underline{c}(2Lh_{\tau_n})^d \tau_n \pi_{\tau_n}}{56}\right)
$$

$$
+ 4N \exp\left(-\frac{c_5^2 \underline{c}(2Lh_{\tau_n})^d \tau_n \pi_{\tau_n}(\epsilon - w(Lh_{\tau_n}; f_i))^2}{8c_4^2 v^2 + 8c_4 c(\epsilon - w(Lh_{\tau_n}; f_i))}\right) \quad \text{(S3.3)}
$$

where, the above inequality follows from Lemma 2 and (S6.2), and the fact that $E(M_{n+1}(X_{n+1}) \mid \mathcal{A}_N) \geq \underline{c}(2Lh_{\tau_n})^d \tau_n$.

Now, we want to remove the condition on $\mathcal{A}_N$ from the conditional probability above. Recall that $d_j \overset{\text{ind}}{\sim} G_j$, for $j \geq 1$. Therefore, for the known visiting times $\{s_j, j \geq 1\}$, $P(t_j \leq n) = P(d_j + s_j \leq n) = P(d_j \leq n - s_j) = G_j(n - s_j)$, hence,

$$
P(|\hat{f}_{i,n+1}(X_{n+1}) - f_i(X_{n+1})| \geq \epsilon)
$$

$$
= P\left(|\hat{f}_{i,n+1}(X_{n+1}) - f_i(X_{n+1})| \geq \epsilon, \tau_n \leq \frac{\sum_{j=1}^n G_j(n - s_j)}{2}\right)
$$

$$
+ P\left(|\hat{f}_{i,n+1}(X_{n+1}) - f_i(X_{n+1})| \geq \epsilon, \tau_n > \frac{\sum_{j=1}^n G_j(n - s_j)}{2}\right)
$$

$$
\leq P\left(\tau_n \leq \frac{\sum_{j=1}^n G_j(n - s_j)}{2}\right)
$$

$$
+ P\left(|\hat{f}_{i,n+1}(X_{n+1}) - f_i(X_{n+1})| \geq \epsilon, \tau_n > \frac{\sum_{j=1}^n G_j(n - s_j)}{2}\right)
$$

$$
\leq P\left(\tau_n \leq \frac{\sum_{j=1}^n G_j(n - s_j)}{2}\right)
$$

$$
+ P\left(|\hat{f}_{i,n+1}(X_{n+1}) - f_i(X_{n+1})| \geq \epsilon, \tau_n > \frac{a_1 q(n)}{2}\right)
$$

$$
= P\left(\tau_n \leq \frac{\sum_{j=1}^n G_j(n - s_j)}{2}\right)
$$

$$
+ E\left[P_{\mathcal{A}_N}\left(|\hat{f}_{i,n+1}(X_{n+1}) - f_i(X_{n+1})| \geq \epsilon, \tau_n > \frac{a_1 q(n)}{2}\right)\right], \quad \text{(S3.4)}
$$

for large enough $n$, where $a_1$ is a positive constant arising from Assumption 7. Also, note that the second term in the last equality (S3.4) is due to the law of iterated expectation. Let $q_1(n) = q(n)/2$.

For $\tau_n > a_1 q_1(n)$, since we have the condition that $h_{q(n)}^d \pi_{q(n)} q(n) / \log n \to \infty$, for large enough n, we can assume that $h_{\tau_n}^d \tau_n \geq \tilde{a}_1 h_{q_1(n)}^d q_1(n)$ and $h_{\tau_n}^d \pi_{\tau_n} \tau_n \geq \tilde{a}_1 h_{q_1(n)}^d \pi_{q_1(n)} q_1(n)$, where $\tilde{a}_1$ is a constant that is function of constant $a_1$, which depends on the user determined choice of sequences $\{\pi_n\}$ and $\{h_n\}$. For large enough n, $\epsilon - w(Lh_{q(n)}; f_i) > 0$, and we have using (S3.3) and (S6.2) in (S3.4),

$$
\leq \exp\left(-\frac{3a_1 q_1(n)}{14}\right) + \exp\left(-\frac{3\underline{c}\tilde{a}_1 (2Lh_{q_1(n)})^d q_1(n)}{28}\right)
$$

$$
+ \exp\left(-\frac{3\underline{c}\tilde{a}_1 (2Lh_{q_1(n)})^d q_1(n)\pi_{q_1(n)}}{56}\right)
$$

$$
+ 4N \exp\left(-\frac{c_5^2 \underline{c}\tilde{a}_1 (2Lh_{q_1(n)})^d q_1(n)\pi_{q_1(n)}(\epsilon - w(Lh_{q_1(n)}; f_i))^2}{8c_4^2 v^2 + 8c_4 c(\epsilon - w(Lh_{q_1(n)}; f_i))}\right)
$$

$$
\leq \exp\left(-\frac{3a_1 q(n)}{28}\right) + \exp\left(-\frac{3\underline{c}\tilde{a}_1 (2Lh_{q(n)})^d q(n)}{56}\right)
$$

$$
+ \exp\left(-\frac{3\underline{c}\tilde{a}_1 (2Lh_{q(n)})^d q(n)\pi_{q(n)}}{112}\right)
$$

$$
+ 4N \exp\left(-\frac{c_5^2 \underline{c}\tilde{a}_1 (2Lh_{q(n)})^d q(n)\pi_{q(n)}(\epsilon - w(Lh_{q(n)}; f_i))^2}{16c_4^2 v^2 + 16c_4 c(\epsilon - w(Lh_{q(n)}; f_i))}\right). \tag{S3.5}
$$

Given $0 < \delta < 1$, we want to bound the right hand side above by $\delta$. To do that for the first three terms, given total time horizon $N$, we define a special time point $n_\delta'$ as in (7) by,

$$
n_\delta' = \min\left\{n > m_0 : \exp\left(-\frac{3\underline{c}\tilde{a}_1 (2Lh_{q(n)})^d \pi_{q(n)} q(n)}{112}\right) \leq \frac{\delta}{4\ell N}\right\}. \tag{S3.6}
$$

For the fourth term in the right hand side of (S3.5), we want to choose an $\epsilon$ such that,

$$
4N \exp\left(-\frac{c_5^2 \underline{c}\tilde{a}_1 (2Lh_{q(n)})^d \pi_{q(n)} q(n)(\epsilon - w(Lh_{q(n)}; f_i))^2}{16c_4^2 v^2 + 16c_4 c(\epsilon - w(Lh_{q(n)}; f_i))}\right) \leq \frac{\delta}{4\ell N},
$$

One such value for $\epsilon$ as shown in Lemma S1 is given by,

$$
\tilde{\epsilon}_{i,n} = w(Lh_{q(n)}; f_i) + \sqrt{\frac{64c_4^2 v^2 \log(16\ell N^2/\delta)}{c_5^2 \underline{c}\tilde{a}_1 (2L)^d h_{q(n)}^d \pi_{q(n)} q(n)}}. \tag{S3.7}
$$

By (S3.5), (S3.6) and (S3.7), for $n \geq n_\delta'$, we have that,

$$
P\left(|\hat{f}_{i,n+1}(X_{n+1}) - f_i(X_{n+1})| \geq \tilde{\epsilon}_{i,n}\right) \leq \frac{\delta}{4\ell N} + \frac{\delta}{4\ell N} + \frac{\delta}{4\ell N} + \frac{\delta}{4\ell N} = \frac{\delta}{\ell N},
$$

which implies that,

$$
P\left(\sum_{n_\delta'+1}^{N} 2 \sup_{1 \leq i \leq \ell} |\hat{f}_{i,n}(X_n) - f_i(X_n)| \geq \sum_{n_\delta'+1}^{N} 2 \max_{1 \leq i \leq \ell} \tilde{\epsilon}_{i,n-1}\right) \leq \delta. \tag{S3.8}
$$

Now we want to get a bound for the randomization error.

Let $\sigma_t = \min\{\bar{n} : \sum_{j=n_\delta'+1}^{\bar{n}} I(t_j \leq N) \geq t\}$, for $t \in \mathbb{Z}$. Recall that for strategy $\eta_2$, we update only when a new reward is observed that is at every $\sigma_t, t \geq 1$. In between the time points corresponding to two consecutive reward observations, $\{\pi_t\}$ takes the same as the value for the previous observed case. In other words, we have $\sigma_{t+1} - \sigma_t$ same values $(\ell-1)\pi_t$ for the exploration probability for each $t$, hence $\sum_{n=n_\delta'+1}^{N} P(I_n \neq \hat{i}_n) = \sum_{n=n_\delta'+1}^{N} (\ell-1)\pi_{\tau_n} = \sum_{t=1}^{\tau_N} (\sigma_{t+1} - \sigma_t)(\ell-1)\pi_t$, and w.l.o.g., assume that $\sigma_{\tau_N+1} = N$.

Given $\epsilon > 0$ and the set of observed indices by time $N$, $\mathcal{A}_N$, we have by the Bernstein's inequality (S6.1) that,

$$P_{\mathcal{A}_N, X^N}\left(A\left(\sum_{n=n'_\delta+1}^N I(I_n \neq \hat{i}_n) - \sum_{t=1}^{\tau_N}(\sigma_{t+1} - \sigma_t)(\ell - 1)\pi_t\right) \geq \epsilon\right)$$

$$\leq \exp\left(-\frac{\epsilon^2}{2A^2(\sum_{t=1}^{\tau_N}(\sigma_{t+1} - \sigma_t)(\ell - 1)\pi_t[1 - (\ell - 1)\pi_t] + \epsilon/3)}\right). \tag{S3.9}$$

Next, for some positive constant $M > 0$, we study the event $B_t := \{\sigma_{t+1} - \sigma_t > M\}$ for $t \geq 1$. Note that, the event $B_t$ is contained in the event that the first $M/2$ cases in $[\sigma_t, \sigma_{t+1}]$ are delayed by more than $M/2$, that is,

$$\{\sigma_{t+1} - \sigma_t > M\} \subset \left\{d_{\sigma_t+1} > \frac{M}{2}, \ldots, d_{\sigma_t+M/2} > \frac{M}{2}\right\}.$$

Therefore, using this fact and by independence of delays, we have that,

$$P(\sigma_{t+1} - \sigma_t > M) \leq P\left(d_{\sigma_t+1} > \frac{M}{2}, \ldots, d_{\sigma_t+M/2} > \frac{M}{2}\right)$$

$$\leq \prod_{s=1}^{M/2} P\left(d_{\sigma_t+s} > \frac{M}{2}\right)$$

$$= \prod_{s=1}^{M/2}\left(1 - G_{d_{\sigma_t+s}}\left(\frac{M}{2}\right)\right) \tag{S3.10}$$

$$\leq \left(\frac{M/2 - \sum_{s=1}^{M/2} G_{d_{\sigma_t+s}}(M/2)}{M/2}\right)^{M/2}$$

$$\leq \left(1 - \frac{a_1 q(M/2)}{M/2}\right)^{M/2}, \quad \text{for all } t = 1, \ldots, \tau_N, \tag{S3.11}$$

where the second to last inequality comes from AM-GM inequality and the last inequality follows from Assumption 7 and $q(M/2) \leq M/2$ for all $M$, by construction. We see that the above upper bound decays at an exponential rate as $M$ grows. As the above right hand side is free of $t$ (by independence of delays), we have that,

$$P\left(\max_t(\sigma_{t+1} - \sigma_t) \geq M\right) \leq \left(1 - \frac{a_1 q(M/2)}{M/2}\right)^{M/2}.$$

We can choose $M$ such that, for a given $\delta$,

$$\left(1 - \frac{a_1 q(M/2)}{M/2}\right)^{M/2} = \delta. \tag{S3.12}$$

Given $q$ and $a_1$, we can solve for $M$ in the above equation. Consequently, since $M$ will depend on $\delta$, we denote it as $M_\delta$. Depending on what $q$ is for a given problem, we will always be able to find a corresponding $M_\delta$.

Also, note that using Hoeffding's inequality (S4), we have that,

$$P\left(\tau_N \geq E(\tau_N) + \frac{\epsilon}{A}\right) \leq \exp\left(-\frac{2\epsilon^2}{A^2 N}\right). \tag{S3.13}$$

We can choose $\epsilon_1(N, \delta) = \sqrt{(N/2)\log(1/\delta)}$ such that this probability is less that $\delta$, that is,

$$P(\tau_N \geq E(\tau_N) + \epsilon_1(N, \delta)) \leq \delta. \tag{S3.14}$$

114 Now consider,

$$P\Big(A\Big(\sum_{n=n'_\delta+1}^N I(I_n \neq \hat{i}_n) - \sum_{t=1}^{\tau_N}(\sigma_{t+1}-\sigma_t)(\ell-1)\pi_t\Big) \geq \epsilon\Big)$$

$$= P\Big(\Big(\sum_{n=n'_\delta+1}^N I(I_n \neq \hat{i}_n) - \sum_{t=1}^{\tau_N}(\sigma_{t+1}-\sigma_t)(\ell-1)\pi_t\Big) \geq \frac{\epsilon}{A}, \max_t(\sigma_{t+1}-\sigma_t) \geq M_\delta\Big)$$

$$+ P\Big(\Big(\sum_{n=n'_\delta+1}^N I(I_n \neq \hat{i}_n) - \sum_{t=1}^{\tau_N}(\sigma_{t+1}-\sigma_t)(\ell-1)\pi_t\Big) \geq \frac{\epsilon}{A}, \max_t(\sigma_{t+1}-\sigma_t) < M_\delta\Big)$$

$$\leq P\Big(\max_t(\sigma_{t+1}-\sigma_t) \geq M_\delta\Big)$$

$$+ P\Big(\Big(\sum_{n=n'_\delta+1}^N I(I_n \neq \hat{i}_n) - \sum_{t=1}^{\tau_N}(\sigma_{t+1}-\sigma_t)(\ell-1)\pi_t\Big) \geq \frac{\epsilon}{A}, \max_t(\sigma_{t+1}-\sigma_t) < M_\delta,$$

$$\tau_N \geq E(\tau_N) + \frac{\epsilon}{A}\Big)$$

$$+ P\Big(\Big(\sum_{n=n'_\delta+1}^N I(I_n \neq \hat{i}_n) - \sum_{t=1}^{\tau_N}(\sigma_{t+1}-\sigma_t)(\ell-1)\pi_t\Big) \geq \frac{\epsilon}{A},$$

$$\max_t(\sigma_{t+1}-\sigma_t) < M_\delta, \tau_N < E(\tau_N) + \frac{\epsilon}{A}\Big)$$

$$\leq P\Big(\max_t(\sigma_{t+1}-\sigma_t) \geq M_\delta\Big) + P\Big(\tau_N \geq E(\tau_N) + \frac{\epsilon}{A}\Big)$$

$$+ P\Big(\Big(\sum_{n=n'_\delta+1}^N I(I_n \neq \hat{i}_n) - \sum_{t=1}^{\tau_N}(\sigma_{t+1}-\sigma_t)(\ell-1)\pi_t\Big) \geq \frac{\epsilon}{A},$$

$$\max_t(\sigma_{t+1}-\sigma_t) < M_\delta, \tau_N < E(\tau_N) + \frac{\epsilon}{A}\Big)$$

$$\leq \delta + \exp\Big(-\frac{2\epsilon^2}{A^2 N}\Big)$$

$$+ E\Big[P_{\mathcal{A}_N, X^N}\Big(\Big(\sum_{n=n'_\delta+1}^N I(I_n \neq \hat{i}_n) - \sum_{t=1}^{\tau_N}(\sigma_{t+1}-\sigma_t)(\ell-1)\pi_t\Big) \geq \frac{\epsilon}{A},$$

$$\max_t(\sigma_{t+1}-\sigma_t) < M_\delta, \tau_N < E(\tau_N) + \frac{\epsilon}{A}\Big)\Big], \qquad \text{(S3.15)}$$

115 where the first term follows from (S3.11) and the definition of $M_\delta$ (S3.12), the second term from
116 (S3.13) and last inequality follows from law of iterated expectation.

117 Then using (S3.9) we have that,

$$P_{\mathcal{A}_N, X^N}\Big(A\Big(\sum_{n=n'_\delta+1}^N I(I_n \neq \hat{i}_n) - \sum_{t=1}^{\tau_N}(\sigma_{t+1}-\sigma_t)(\ell-1)\pi_t\Big) \geq \epsilon,$$

$$\max_t(\sigma_{t+1}-\sigma_t) < M_\delta, \tau_N < E(\tau_N) + \frac{\epsilon}{A}\Big)$$

$$\leq \begin{cases} \exp\Big(-\dfrac{\epsilon^2}{2A^2 M_\delta(\mathrm{E}(\tau_N)+\epsilon)/4 + \epsilon/3}\Big), & \text{if } \max_t(\sigma_{t+1}-\sigma_t) < M_\delta, \\ & \hspace{3em} \tau_N < \mathrm{E}(\tau_N) + \epsilon/A; \\ 0, & \text{otherwise.} \end{cases}$$

118    Using this in (S3.15), we get,

$$
\mathrm{E}P_{\mathcal{A}_N, X^N}\left( A\left( \sum_{n=n'_\delta+1}^{N} I(I_n \neq \hat{i}_n) - \sum_{t=1}^{\tau_N}(\sigma_{t+1} - \sigma_t)(\ell - 1)\pi_t \right) \geq \epsilon, \right.
$$

$$
\left. \max_t(\sigma_{t+1} - \sigma_t) \leq M_\delta, \tau_N < E(\tau_N) + \epsilon/A \right)
$$

$$
\leq \exp\left( -\frac{\epsilon^2}{2A^2 M_\delta(\mathrm{E}(\tau_N) + \epsilon)/4 + \epsilon/3} \right).
\tag{S3.16}
$$

119    Therefore, combining (S3.15) and (S3.16), we get that with probability at least 1-$\delta$,

$$
P\left( A\left( \sum_{n=n'_\delta+1}^{N} I(I_n \neq \hat{i}_n) - \sum_{t=1}^{\tau_N}(\sigma_{t+1} - \sigma_t)(\ell - 1)\pi_t \right) \geq \epsilon \right)
$$

$$
\leq \delta + \exp\left( -\frac{2\epsilon^2}{A^2 N} \right) + \exp\left( -\frac{\epsilon^2}{2A^2 M_\delta(\mathrm{E}(\tau_N) + \epsilon)/4 + \epsilon/3} \right).
$$

In order to bound the right hand side by 2$\delta$, let,

$$
\epsilon_{N,\delta} = \max\left\{ A\sqrt{M_\delta \frac{E(\tau_N)}{2} \log\left(\frac{2}{\delta}\right)}, A\sqrt{\frac{N}{2} \log\left(\frac{2}{\delta}\right)} \right\}.
$$

120    For this chosen $\epsilon$, we have that,

$$
P\left( A\left( \sum_{n=n'_\delta+1}^{N} I(I_n \neq \hat{i}_n) - \sum_{t=1}^{\tau_N}(\sigma_{t+1} - \sigma_t)(\ell - 1)\pi_t \right) \geq \epsilon_{N,\delta} \right) \leq 2\delta
$$

$$
\Rightarrow P\left( A\sum_{n=n'_\delta+1}^{N} I(I_n \neq \hat{i}_n) \geq A\sum_{t=1}^{\tau_N}(\sigma_{t+1} - \sigma_t)(\ell - 1)\pi_t + \epsilon_{N,\delta} \right) \leq 2\delta.
\tag{S3.17}
$$

121    Note that,

$$
P\left( A\left( \sum_{n=n'_\delta+1}^{N} I(I_n \neq \hat{i}_n) - \sum_{t=1}^{\tau_N}(\sigma_{t+1} - \sigma_t)(\ell - 1)\pi_t \right) \geq \epsilon_{N,\delta} \right)
$$

$$
\geq P\left( A\left( \sum_{n=n'_\delta+1}^{N} I(I_n \neq \hat{i}_n) - \sum_{t=1}^{\tau_N}(\sigma_{t+1} - \sigma_t)(\ell - 1)\pi_t \right) \geq \epsilon_{N,\delta}, \right.
$$

$$
\left. \tau_N \leq \mathrm{E}(\tau_N) + \epsilon_1(N,\delta), \max_t(\sigma_{t+1} - \sigma_t) \leq M_\delta \right)
$$

$$
= P\left( A\left( \sum_{n=n'_\delta+1}^{N} I(I_n \neq \hat{i}_n) - \sum_{t=1}^{\tau_N}(\sigma_{t+1} - \sigma_t)(\ell - 1)\pi_t \right) \geq \epsilon_{N,\delta} \middle| \tau_N \leq \mathrm{E}(\tau_N) \right.
$$

$$
\left. + \epsilon_1(N,\delta), \max_t(\sigma_{t+1} - \sigma_t) \leq M_\delta \right) \times
$$

$$
P\left( \tau_N \leq \mathrm{E}(\tau_N) + \epsilon_1(N,\delta) \right) P\left( \max_t(\sigma_{t+1} - \sigma_t) \leq M_\delta \right)
$$

$$
\geq P\left( A\left( \sum_{n=n'_\delta+1}^{N} I(I_n \neq \hat{i}_n) - \sum_{t=1}^{\mathrm{E}(\tau_N)+\epsilon_1(N,\delta)} M_\delta(\ell - 1)\pi_t \right) \geq \epsilon_{N,\delta} \right)(1 - \delta)^2,
\tag{S3.18}
$$

where the last inequality follows from (S3.12) and (S3.14). Now, from (S3.17) and (S3.18), we get,

$$P\left(A\left(\sum_{n=n_\delta'+1}^{N}I(I_n\neq\hat{i}_n)-\sum_{t=1}^{\mathrm{E}(\tau_N)+\epsilon_1(N,\delta)}M_\delta(\ell-1)\pi_t\right)\geq\epsilon_{N,\delta}\right)(1-\delta)^2$$

$$\leq P\left(A\left(\sum_{n=n_\delta'+1}^{N}I(I_n\neq\hat{i}_n)-\sum_{t=1}^{\tau_N}(\sigma_{t+1}-\sigma_t)(\ell-1)\pi_t\right)\geq\epsilon_{N,\delta}\right)\qquad\text{(S3.19)}$$

$$\leq 2\delta$$

$$\Rightarrow P\left(A\left(\sum_{n=n_\delta'+1}^{N}I(I_n\neq\hat{i}_n)-\sum_{t=1}^{\mathrm{E}(\tau_N)+\epsilon_1(N,\delta)}M_\delta(\ell-1)\pi_t\right)\geq\epsilon_{N,\delta}\right)\leq\frac{2\delta}{(1-\delta)^2}\qquad\text{(S3.20)}$$

From (S3.8) and (S3.20), we get that with probability at least $1-2\delta/(1-\delta)^2$, the cumulative regret for strategy $\eta_2$ satisfies,

$$R_N(\eta_2)<An_\delta'+\sum_{n=n_\delta'+1}^{N}2\left(\max_{1\leq i\leq\ell}w(Lh_{q(n)};f_i)+\sqrt{\frac{64c_4^2v^2\log(12\ell N^2/\delta)}{c_5^2\underline{c}(2L)^dh_{q(n)}^d\pi_{q(n)}q(n)}}\right)$$

$$+A\sum_{t=1}^{N^*(\delta)}M_\delta(\ell-1)\pi_t+\max\left\{A\sqrt{M_\delta\frac{E(\tau_N)}{2}\log\left(\frac{2}{\delta}\right)},A\sqrt{\left(\frac{N}{2}\right)\log\left(\frac{2}{\delta}\right)}\right\},$$

for $N^*(\delta)=\mathrm{E}(\tau_N)+\epsilon_1(N,\delta)$. Let $\delta<1/4$ and we get the desired result. $\qquad\square$

## S4   Proof of Theorem 2

*Proof of Theorem 2.* Similar to Theorem 1, we will first find an upper bound for the estimation error. In order to do so, in (6) of Lemma 3, we first remove condition on $X^n$ and then remove the condition on $\mathcal{A}_N$ from the conditional probability statement of the Lemma. Given arm $i$, and $n$ large enough such that, $n\geq m_0+1$ and $\epsilon>w(Lh_{\tau_n};f_i)$ a.s. (such $n$ exists from Lemma 1), consider,

$$P_{\mathcal{A}_N}\left(|\hat{f}_{i,n+1}-f_i(X_{n+1})|\geq\epsilon\right)$$

$$\leq P_{\mathcal{A}_N}\left(|\hat{f}_{i,n+1}-f_i(X_{n+1})|\geq\epsilon,M_{n+1}(X_{n+1})\leq\frac{\underline{c}(2Lh_{\tau_n})^d\tau_n}{2}\right)$$

$$\qquad+P_{\mathcal{A}_N}\left(|\hat{f}_{i,n+1}-f_i(X_{n+1})|\geq\epsilon,M_{n+1}(X_{n+1})>\frac{\underline{c}(2Lh_{\tau_n})^d\tau_n}{2}\right)$$

$$\leq P_{\mathcal{A}_N}\left(M_{n+1}(X_{n+1})\leq\frac{\underline{c}(2Lh_{\tau_n})^d\tau_n}{2}\right)$$

$$\qquad+P_{\mathcal{A}_N}\left(|\hat{f}_{i,n+1}-f_i(X_{n+1})|\geq\epsilon,M_{n+1}(X_{n+1})>\frac{\underline{c}(2Lh_{\tau_n})^d\tau_n}{2}\right)$$

$$\leq\exp\left(-\frac{3\underline{c}(2Lh_{\tau_n})^d\tau_n}{28}\right)+\exp\left(-\frac{3\underline{c}(2Lh_{\tau_n})^d\tau_n\pi_n}{56}\right)$$

$$\qquad+4N\exp\left(-\frac{c_5^2\underline{c}(2Lh_{\tau_n})^d\tau_n\pi_n(\epsilon-w(Lh_{\tau_n};f_i))^2}{8c_4^2v^2+8c_4c(\epsilon-w(Lh_{\tau_n};f_i))}\right),\qquad\text{(S4.1)}$$

where, the above inequality follows from Lemma 3 and (S6.2).

Now, we want to remove the condition on $\mathcal{A}_N$ from the above conditional probability inequality. Recall that $d_j\overset{\text{ind}}{\sim}G_j$, for $j\geq1$. Therefore, for the known visiting times $\{s_j,j\geq1\}$, $P(t_j\leq n)=$

135    $P(d_j + s_j \leq n) = P(d_j \leq n - s_j) = G_j(n - s_j)$, and hence,

$$P\left(|\hat{f}_{i,n+1}(X_{n+1}) - f_i(X_{n+1})| \geq \epsilon\right)$$

$$= P\left(|\hat{f}_{i,n+1}(X_{n+1}) - f_i(X_{n+1})| \geq \epsilon, \tau_n \leq \frac{\sum_{j=1}^n G_j(n - s_j)}{2}\right)$$

$$+ P\left(|\hat{f}_{i,n+1}(X_{n+1}) - f_i(X_{n+1})| \geq \epsilon, \tau_n > \frac{\sum_{j=1}^n G_j(n - s_j)}{2}\right)$$

$$\leq P\left(\tau_n \leq \frac{\sum_{j=1}^n G_j(n - s_j)}{2}\right) + P\left(|\hat{f}_{i,n+1}(X_{n+1}) - f_i(X_{n+1})| \geq \epsilon,\right.$$

$$\left. \tau_n > \frac{\sum_{j=1}^n G_j(n - s_j)}{2}\right)$$

$$\leq P\left(\tau_n \leq \frac{\sum_{j=1}^n G_j(n - s_j)}{2}\right) + \mathrm{E}P_{\mathcal{A}_N}\left(|\hat{f}_{i,n+1}(X_{n+1}) - f_i(X_{n+1})| \geq \epsilon,\right.$$

$$\left. \tau_n > \frac{a_1 q(n)}{2}\right),$$

136    where $\sum_{j=1}^n G_j(n - s_j) = \Omega(q(n))$ from Assumption 7, that is, for large enough $n$, we would have
137    that $\sum_{j=1}^n G_j(n - s_j) \geq a_1 q(n)$ for some positive constant $a_1$. Let $q_1(n) = a_1 q(n)/2$, we get, for
138    $\tau_n > q_1(n)$, since we have the condition that $h_{q(n)}^d \pi_n q(n)/\log n \to \infty$, for large enough n, we
139    can assume that $h_{\tau_n}^d \tau_n \geq \tilde{\tilde{a}}_1 h_{q_1(n)}^d q_1(n)$ and $h_{\tau_n}^d \pi_n \tau_n \geq \tilde{\tilde{a}}_1 h_{q_1(n)}^d \pi_n q_1(n)$, where $\tilde{\tilde{a}}_1$ is a positive
140    constant depending on $a_1$ and the choice of hyperparameter sequences $\{h_n\}$ and $\{\pi_n\}$. For large
141    enough $n$, we have that $\epsilon - w(Lh_{q(n)}; f_i) > 0$. Now, using (S4.1) and (S6.2), we get,

$$\leq \exp\left(-\frac{3q_1(n)}{14}\right) + \exp\left(-\frac{3\underline{c}(2Lh_{q_1(n)})^d q_1(n)}{28}\right)$$

$$+ \exp\left(-\frac{3\underline{c}(2Lh_{q_1(n)})^d q_1(n)\pi_n}{56}\right)$$

$$+ 4N \exp\left(-\frac{c_5^2 \underline{c}(2Lh_{q_1(n)})^d q_1(n)\pi_n(\epsilon - w(Lh_{q_1(n)}; f_i))^2}{8c_4^2 v^2 + 8c_4 c(\epsilon - w(Lh_{q_1(n)}; f_i))}\right)$$

$$\leq \exp\left(-\frac{3a_1 q(n)}{28}\right) + \exp\left(-\frac{3\underline{c}\tilde{\tilde{a}}_1(2Lh_{q(n)})^d q(n)}{56}\right)$$

$$+ \exp\left(-\frac{3\underline{c}\tilde{\tilde{a}}_1(2Lh_{q(n)})^d q(n)\pi_n}{112}\right)$$

$$+ 4N \exp\left(-\frac{c_5^2 \underline{c}\tilde{\tilde{a}}_1(2Lh_{q(n)})^d q(n)\pi_n(\epsilon - w(Lh_{q(n)}; f_i))^2}{16c_4^2 v^2 + 16c_4 c(\epsilon - w(Lh_{q(n)}; f_i))}\right). \tag{S4.2}$$

142    Given $0 < \delta < 1$, we want to bound the R.H.S. above by $\delta$. To do that for the first three terms, given
143    total time horizon $N$, we define a special time point $n_\delta''$ as in (9),

$$n_\delta'' = \min\left\{n > m_0 : \exp\left(-\frac{3\underline{c}\tilde{\tilde{a}}_1(2Lh_{q(n)})^d \pi_n q(n)}{112}\right) \leq \frac{\delta}{4\ell N}\right\}. \tag{S4.3}$$

144    For the fourth term in the R.H.S. of (S4.2), we want to choose an $\epsilon$ such that,

$$4N \exp\left(-\frac{c_5^2 \underline{c}\tilde{\tilde{a}}_1(2Lh_{q(n)})^d \pi_n q(n)(\epsilon - w(Lh_{q(n)}; f_i))^2}{16c_4^2 v^2 + 16c_4 c(\epsilon - w(Lh_{q(n)}; f_i))}\right) \leq \frac{\delta}{4\ell N}.$$

One such value for $\epsilon$ is given in Lemma S2 as,

$$\tilde{\epsilon}'_{i,n} = w(Lh_{q(n)}; f_i) + \sqrt{\frac{64c_4^2 v^2 \log(16\ell N^2/\delta)}{c_5^2 \underline{c}\tilde{a}_1(2L)^d h_{q(n)}^d \pi_n q(n)}}.$$ (S4.4)

By (S4.2), (S4.3) and (S4.4), for $n \geq n''_\delta$, we have that,

$$P\left(|\hat{f}_{i,n+1}(X_{n+1}) - f_i(X_{n+1})| \geq \tilde{\epsilon}'_{i,n}\right) \leq \frac{\delta}{4\ell N} + \frac{\delta}{4\ell N} + \frac{\delta}{4\ell N} + \frac{\delta}{4\ell N} = \frac{\delta}{\ell N},$$

which implies that,

$$P\left(\sum_{n''_\delta+1}^{N} 2 \sup_{1 \leq i \leq \ell} |\hat{f}_{i,n}(X_n) - f_i(X_n)| \geq \sum_{n''_\delta+1}^{N} 2 \max_{1 \leq i \leq \ell} \tilde{\epsilon}'_{i,n-1}\right) \leq \delta.$$ (S4.5)

Now we want to get a bound for the randomization error regret. Given $\epsilon > 0$, since $P(I_n \neq \hat{i}_n) = (\ell - 1)\pi_n$, we have by the Hoeffding's inequality that,

$$P\left(A\left(\sum_{n=n''_\delta+1}^{N} I(I_n \neq \hat{i}_n) - \sum_{n=n''_\delta+1}^{N} (\ell - 1)\pi_n\right) \geq \epsilon\right) \leq \exp\left(-\frac{2\epsilon^2}{NA^2}\right).$$

Take $\epsilon = A\sqrt{N/2 \log(1/\delta)}$, we get,

$$P\left(A \sum_{n=n''_\delta+1}^{N} I(I_n \neq \hat{i}_n) \geq A \sum_{n=n''_\delta+1}^{N} (\ell - 1)\pi_n + A\sqrt{\frac{N}{2} \log\left(\frac{1}{\delta}\right)}\right) \leq \delta.$$ (S4.6)

Therefore, from (S4.5) and (S4.6), we get that with probability at least $1 - 2\delta$, the cumulative regret satisfies,

$$R_N(\eta_1) < An''_\delta + \sum_{n=n''_\delta+1}^{N} 2\left(\max_{1 \leq i \leq \ell} w(Lh_{q(n)}; f_i) + \frac{C_{N,\delta}}{\sqrt{h_{q(n)}^d \pi_n q(n)}} + A(\ell - 1)\pi_n\right)$$
$$+ A\sqrt{\left(\frac{N}{2} \log\left(\frac{1}{\delta}\right)\right)},$$

where $C_{N,\delta} = \sqrt{64c_4^2 v^2 \log(12\ell N^2/\delta)/c_5^2 \underline{c}\tilde{a}_1(2L)^d}$. Hence the desired result. $\square$

*Proof of Theorem S1.* Since a lot of steps remain the same as Theorems 1 and 2, we outline the steps that change here. Firstly, in lemma S2.1, recall,

$$P_{X^n, \mathcal{A}_N}\left(|\hat{f}_{i,n+1}(x) - f_i(x)| \geq \epsilon\right)$$
$$\overset{a}{\leq} \exp\left(-\frac{3M_{n+1}(x)\pi_{\tau_n}}{28}\right) + P_{X^n, \mathcal{A}_N}\left(|\hat{f}_{i,n+1}(x) - f_i(x)| \geq \epsilon, \frac{M_{i,n+1}(x)}{M_{n+1}(x)} > \frac{\pi_{\tau_n}}{2}, B_{i,n}\right)$$
$$+ P_{X^n, \mathcal{A}_N}\left(|\hat{f}_{i,n+1}(x) - f_i(x)| \geq \epsilon, \frac{M_{i,n+1}(x)}{M_{n+1}(x)} > \frac{\pi_{\tau_n}}{2}, B_{i,n}^c\right)$$
$$=: \exp\left(-\frac{3M_{n+1}(x)\pi_{\tau_n}}{28}\right) + A_1 + A_2.$$

For $A_1$, by applying using lemma S9, (S2.3) will become,

$$A_1 \leq \begin{cases} 2N \exp\left(-\dfrac{c_5^2 M_{n+1}(x)\pi_{\tau_n}(\epsilon - w(Lh_{\tau_n}; f_i))^2}{4C^2 v^2}\right) & \text{if } 0 < \epsilon - w(Lh_{\tau_n}; f_i) < v^2 C/\alpha \\ 2N \exp\left(-\dfrac{c_5 M_{n+1}(x)\pi_{\tau_n}(\epsilon - w(Lh_{\tau_n}; f_i))}{4C\alpha}\right) & \text{if } \epsilon - w(Lh_{\tau_n}; f_i) > v^2 C/\alpha. \end{cases}$$

157 Similarly,

$$A_2 \leq \begin{cases} 2N \exp\left(-\dfrac{M_{n+1}(x)\pi_{\tau_n}(\epsilon - w(Lh_{\tau_n}; f_i))^2}{4C^2\nu^2}\right) & \text{if } 0 < \epsilon - w(Lh_{\tau_n}; f_i) < \nu^2 C/\alpha \\ 2N \exp\left(-\dfrac{M_{n+1}(x)\pi_{\tau_n}(\epsilon - w(Lh_{\tau_n}; f_i))}{4C\alpha}\right) & \text{if } \epsilon - w(Lh_{\tau_n}; f_i) > \nu^2 C/\alpha. \end{cases}$$

158 Therefore, Lemma 2 gets modified to the following,

$$P_{X^n, \mathcal{A}_N}\left(|\hat{f}_{i,n+1}(x) - f_i(x)| \geq \epsilon\right) \tag{S4.7}$$

$$\leq \begin{cases} \begin{aligned} &\exp\left(-\dfrac{3M_{n+1}(x)\pi_{\tau_n}}{28}\right) \\ &\quad + 4N \exp\left(-\dfrac{c_5^2 M_{n+1}(x)\pi_{\tau_n}(\epsilon - w(Lh_{\tau_n}; f_i))^2}{4C^2\nu^2}\right), \end{aligned} & \text{if } 0 < \epsilon - w(Lh_{\tau_n}; f_i) < \nu^2 C/\alpha \\ \begin{aligned} &\exp\left(-\dfrac{3M_{n+1}(x)\pi_{\tau_n}}{28}\right) \\ &\quad + 4N \exp\left(-\dfrac{c_5 M_{n+1}(x)\pi_{\tau_n}(\epsilon - w(Lh_{\tau_n}; f_i))}{4C\alpha}\right), \end{aligned} & \text{if } \epsilon - w(Lh_{\tau_n}; f_i) > \nu^2 C/\alpha. \end{cases}$$

159 Following through with the same logic, we get that (S3.5) in proof of 1 would become, for large
160 enough $n$,

$$P(|\hat{f}_{i,n+1}(X_{n+1}) - f_i(X_{n+1})| \geq \epsilon)$$

$$\leq \begin{cases} \begin{aligned} &\exp\left(-\dfrac{3a_1 q(n)}{28}\right) + \exp\left(-\dfrac{3\underline{c}\tilde{a}_1(2Lh_{q(n)})^d q(n)}{56}\right) + \exp\left(-\dfrac{3\underline{c}\tilde{a}_1(2Lh_{q(n)})^d q(n)\pi_{q(n)}}{112}\right) \\ &\quad + 4N \exp\left(-\dfrac{c_5^2 \underline{c}\tilde{a}_1(2Lh_{q(n)})^d q(n)\pi_{q(n)}(\epsilon - w(Lh_{q(n)}; f_i))^2}{8C^2\nu^2}\right), \end{aligned} & \text{if } \epsilon - w(Lh_{q(n)}; f_i) < \nu^2 C/\alpha \\ \begin{aligned} &\exp\left(-\dfrac{3a_1 q(n)}{28}\right) + \exp\left(-\dfrac{3\underline{c}\tilde{a}_1(2Lh_{q(n)})^d q(n)}{56}\right) + \exp\left(-\dfrac{3\underline{c}\tilde{a}_1(2Lh_{q(n)})^d q(n)\pi_{q(n)}}{112}\right) \\ &\quad + 4N \exp\left(-\dfrac{c_5^2 \underline{c}\tilde{a}_1(2Lh_{q(n)})^d q(n)\pi_{q(n)}(\epsilon - w(Lh_{q(n)}; f_i))}{8C\alpha}\right), \end{aligned} & \text{if } \epsilon - w(Lh_{q(n)}; f_i) > \nu^2 C/\alpha \end{cases}$$

161 Then, bounding the above terms by $\delta > 0$, we would get a version of Lemma S1,

$$Z := \tilde{\epsilon}_{i,n} - w(Lh_{q(n)}; f_i)$$

$$= \begin{cases} \sqrt{\dfrac{8C^2\nu^2 \log(16\ell N^2/\delta)}{c_5^2 \underline{c}\tilde{a}_1(2L)^d h_{q(n)}^d \pi_{q(n)} q(n)}} & \text{if } Z < \nu^2 C/\alpha \\ \dfrac{8C\alpha \log(16\ell N^2/\delta)}{c_5^2 \underline{c}\tilde{a}_1(2L)^d h_{q(n)}^d \pi_{q(n)} q(n)} & \text{if } Z > \nu^2 C/\alpha \end{cases}$$

162 The above conditions then imply, case one $Z < \nu^2 C/\alpha$ is the same as,

$$h_{q(n)}^d q(n)\pi_{q(n)} > \frac{8 \log(16\ell N^2/\delta)\alpha^2}{\nu^2 c_5 \underline{c}\tilde{a}_1(2L)^d},$$

163 while case 2, that is, $Z > \nu^2 C/\alpha$ is the compliment of this,

$$h_{q(n)}^d q(n)\pi_{q(n)} < \frac{8 \log(16\ell N^2/\delta)\alpha^2}{\nu^2 c_5 \underline{c}\tilde{a}_1(2L)^d}.$$

164 Note that the modification of sub-exponential errors does not effect the randomization error, we get
165 the final result for 1 as follows, Then for $0 < \delta \leq 1/4$, we have that, with probability at least $1 - \frac{32\delta}{9}$,

the cumulative regret for $\eta_2$ satisfies,

$$
R_N(\eta_2) < \begin{cases}
An'_\delta + \sum_{n=n'_\delta+1}^{N} 2 \left( \max_{1\leq i \leq \ell} w(Lh_{q(n)}; f_i) + \dfrac{C'_{N,\delta}}{\sqrt{h^d_{q(n)}\pi_{q(n)}q(n)}} \right) \\
\quad + A\sum_{t=1}^{N^*(\delta)} M_\delta(\ell-1)\pi_t + \max\left\{ A\sqrt{M_\delta \dfrac{E(\tau_N)}{2} \log\left(\frac{2}{\delta}\right)}, A\sqrt{\left(\frac{N}{2}\right)\log\left(\frac{2}{\delta}\right)} \right\}, \\
\qquad\qquad\qquad \text{if } h^d_{q(n)}q(n)\pi_{q(n)} > \dfrac{8\log(16\ell N^2/\delta)\alpha^2}{\nu^2 c_5 \underline{c}\tilde{a}_1(2L)^d}, \\[4pt]
An'_\delta + \sum_{n=n'_\delta+1}^{N} 2 \left( \max_{1\leq i \leq \ell} w(Lh_{q(n)}; f_i) + \dfrac{C''_{N,\delta}}{h^d_{q(n)}\pi_{q(n)}q(n)} \right) \\
\quad + A\sum_{t=1}^{N^*(\delta)} M_\delta(\ell-1)\pi_t + \max\left\{ A\sqrt{M_\delta \dfrac{E(\tau_N)}{2} \log\left(\frac{2}{\delta}\right)}, A\sqrt{\left(\frac{N}{2}\right)\log\left(\frac{2}{\delta}\right)} \right\}, \\
\qquad\qquad\qquad \text{if } h^d_{q(n)}q(n)\pi_{q(n)} < \dfrac{8\log(16\ell N^2/\delta)\alpha^2}{\nu^2 c_5 \underline{c}\tilde{a}_1(2L)^d},
\end{cases}
$$

where, $\quad C'_{N,\delta} \quad = \quad \sqrt{8C^2\nu^2 \log(16\ell N^2/\delta)/c_5\underline{c}\tilde{a}_1(2L)^d} \quad$ and $\quad C''_{N,\delta} \quad = \quad 8\alpha C\log(16\ell N^2/\delta)/(c_5\underline{c}\tilde{a}_1(2L)^d)$. $\qquad\square$

## S5 More simulation results

Here, we display plots for the three simulation settings for different combinations of hyperparameter sequences, $\{\pi_n = (\log n)^{-2}, h_n = n^{-1/4}\}$ in figure 1 and $\{\pi_n = (\log n)^{-2}, h_n = (\log n)^{-1}\}$ in figure 2, respectively. Again, we used $\lambda_1 = 1$ for strategy $\eta_{\text{adap}_1}$ for all simulation setups and also, $\lambda_2 = 1$ for both setups 2 and 3, but $\lambda_2 = 3$ for $\eta_{\text{adap}_2}$ in setup 1. A thorough investigation may be needed for the selection of $\lambda_1$ and $\lambda_2$ for easy applicability in practical real-world decision making problems. In our simulation study, we get promising results from these adaptive strategies as they perform better (or at par) than both $\eta_1$ and $\eta_2$.

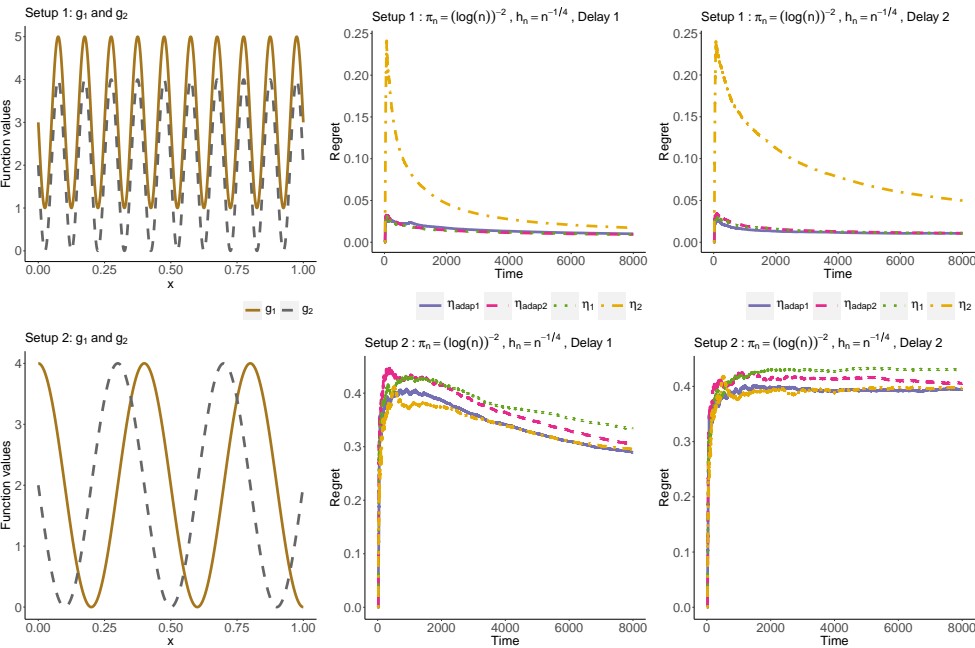

Figure 1: Strategy $\eta_{\text{adap}_1}$ and $\eta_{\text{adap}_2}$ have lower (or at par) cumulative average regret than $\eta_1$ and $\eta_2$ for the three simulation settings.

We also consider another extreme setup, where one of the functions has a big spike and the other is constant.

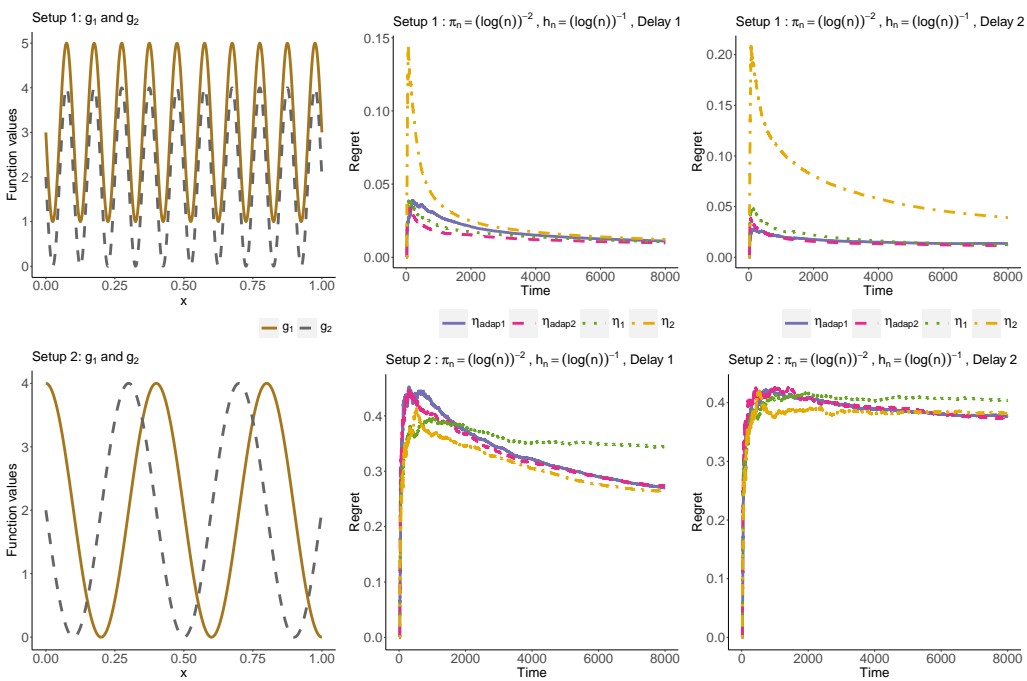

Figure 2: Strategy $\eta_{\text{adap}_1}$ and $\eta_{\text{adap}_2}$ have lower (or at par) cumulative average regret than $\eta_1$ and $\eta_2$ for the two simulation settings.

**Setup 3:** Consider a setup where one arm dominates over majority of the covariate space, except for a small area where it incurs a considerably high regret.

$$g_1(x) = 1, \text{ for all } x \in [0,1]; g_2(x) = \begin{cases} 0 & 0 \le x < 0.5, 0.505 \le x \le 1 \\ 100000x - 50000 & 0.5 \le x < 0.502 \\ 200 & 0.502 \le x < 0.503 \\ -100000 * x + 50500 & 0.503 \le x < 0.505. \end{cases}$$

We look at both the setup $d = 2$, when $f_1(x_1, x_2) = g_1(x_1) * x_2$ and $f_2(x_1, x_2) = g_2(x_1) * x_2$. The corresponding regret plots are in Figure 3. Note that the adaptive strategies $\eta_{\text{adap}_1}$ and $\eta_{\text{adap}_2}$ outperform $\eta_1$ and $\eta_2$ in this setting.

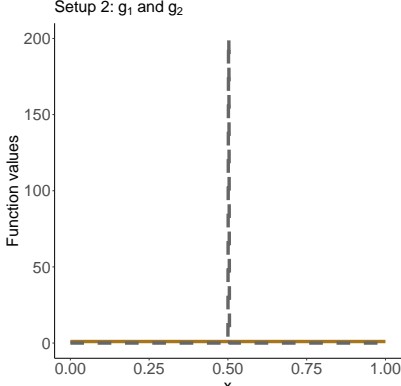

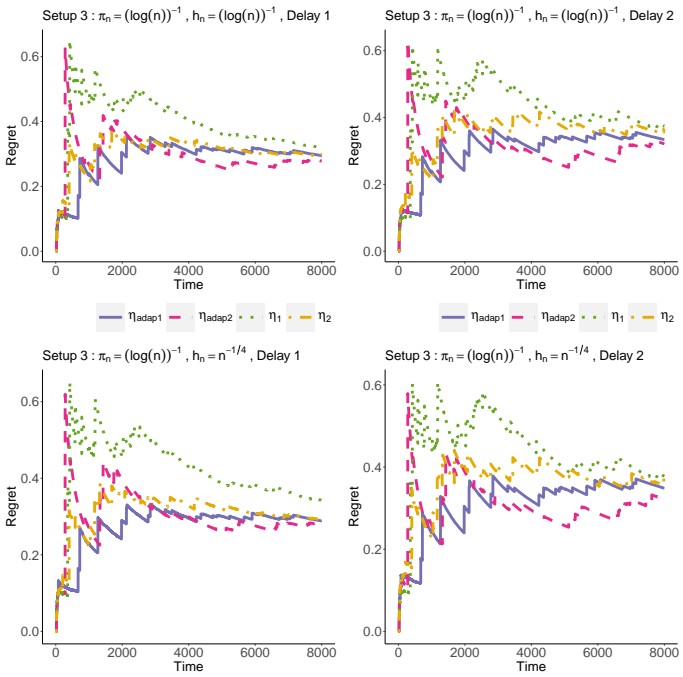

Figure 3: Strategy $\eta_{\text{adap}_1}$ and $\eta_{\text{adap}_2}$ have lower (or at par) cumulative average regret than $\eta_1$ and $\eta_2$ for Setting 3.

## S6   Appendix

In this section, we enlist some well-known technical tools that are used in the paper. We first state the famous Borel-Cantelli Lemma.

**Lemma S3** (A.1). *[Borel-Cantelli] Let $(A_1, A_2, \ldots)$ be a sequence of events in a common probability space $(\Omega, \mathcal{F}, P)$ and set $A = \limsup_{n \to \infty} A_n$. If $\sum_{n=1}^{\infty} P(A_n) < \infty$, then $P(A) = 0$.*

This result is useful in assessing almost sure convergence and is often used in the analysis presented in the following chapters. Next, we define the modulus of continuity, which quantifies the maximum differences in functional values for a given function on a given domain.

**Definition 1.** *Let $x_1, x_2 \in [0,1]^d$. Then $w(h; f)$ denotes a modulus of continuity defined by, $w(h; f) = \sup\{|f(x_1) - f(x_2)| : |x_{1k} - x_{2k}| \le h \text{ for all } 1 \le k \le d\}$.*

It can be seen that if $f$ is continuous then $w(h; f) \to 0$ as $h \to 0$.

Next, we review some concentration inequalities, which are quite standard results and will be used in the following chapters.

### S6.1   Concentration inequalities

**Lemma S4** (Hoeffding's Inequality). *Let $X_1, X_2, \ldots, X_n$ be independent real-valued random variables such that for each $i = 1, \ldots, n$ there exists some $a_i \le b_i$ such that $P[a_i \le X_i \le b_i] = 1$. Then for every $\epsilon > 0$,*

$$P\left[\sum_{i=1}^{n} X_i - E \sum_{i=1}^{n} X_i > \epsilon\right] \le \exp\left(-\frac{2\epsilon^2}{\sum_{i=1}^{n}(b_i - a_i)^2}\right)$$

*More such inequalities with their proofs can be found in Hoeffding (1994).*

The martingale version of Hoeffding inequality has also been derived and is known as the Azuma-Hoeffding inequality.

**Lemma S5** (Azuma-Hoeffding Inequality). *Suppose $\mathcal{F}_j, j = 1, 2, \ldots$ is an increasing filtration of $\sigma$-fields. For each $j \geq 1$, let $X_j$ be $\mathcal{F}_j$-measurable such that $X_j \geq 0$ almost surely, and $a_j \leq X_j \leq b_j$, then for all $\epsilon > 0$, we have,*

$$P\left[\sum_{j=1}^{n} X_j - \sum_{j=1}^{n} E(X_j \mid \mathcal{F}_{j-1}) > \epsilon\right] \leq \exp\left(-\frac{2\epsilon^2}{\sum_{j=1}^{n}(b_j - a_j)^2}\right)$$

One if referred to McDiarmid (1998) for more details and a proof of the inequality.

**Lemma S6.** *A.4[Bernstein's Inequality] Let $X_1, \ldots, X_n$ be independent real-valued random variables with zero mean, and assume that $X_1 \leq 1$ with probability 1. Let $V_j = \text{Var}(X_j)$ and $\sigma^2 = \sum_{j=1}^{n} V_j$. For any $\epsilon > 0$,*

$$P\left[\frac{1}{n}\sum_{i=1}^{n} X_i > \epsilon\right] \leq \exp\left(-\frac{n\epsilon^2}{2\sigma^2 + 2\epsilon/3}\right) \tag{S6.1}$$

Proofs of these inequalities can be found in Cesa-Bianchi and Lugosi (2006).

**Corollary S1.** *Suppose $\tilde{W}_1, \tilde{W}_2, \ldots, \tilde{W}_n$, are independent Bernoulli random variables with success probability $\beta_j$. By Bernstein's inequality in* (S6.1),

$$P\left(\sum_{j=1}^{n} \tilde{W}_j \leq (\sum_{j=1}^{n} \beta_j)/2\right) \leq \exp\left(-\frac{3\sum_{j=1}^{n} \beta_j}{28}\right).$$

The proof follows by substituting $\epsilon = (\sum_{j=1}^{n} \beta_j)/2$ and $X_j = \beta_j - \tilde{W}_j$ in (S6.1). Note that the same inequality holds for any Bernoulli random variable where $W_j$ takes values $a_j \leq 1$, $\forall j \geq 1$ and 0.

The Bernstein's inequality has been extended to the case of martingales.

**Lemma S7** (Bernstein's Inequality for Martingales). *Let $(\Omega, \mathcal{F}, P)$ be a probability space. Let $\mathcal{F}_j, j = 1, 2, \ldots$, be an increasing filtration of sub-$\sigma$-fields of $\mathcal{F}$. Let $X_1, X_2, \ldots$ be random variables on $(\Omega, \mathcal{F}, P)$, such that $X_j$ is $\mathcal{F}_j$-measurable. Assume $|X_j| \leq K$ with probability 1, for all $j \geq 1$. Let $V_j = \text{Var}(X_j \mid \mathcal{F}_{j-1})$ and denote the sum of conditional variances by, Then for all positive real numbers $\epsilon$ and $v$,*

$$P\left(\sum_{j=1}^{n}(X_j - E(X_j|\mathcal{F}_{j-1})) > \epsilon, \sum_{j=1}^{n} V_j \leq v\right) \leq \exp\left(-\frac{\epsilon^2}{2(v + K\epsilon/3)}\right)$$

The proof of this inequality can be found in Freedman (1975).

**Corollary S2** (Extended Bernstein Inequality). *Suppose $\{\mathcal{F}_j, j = 1, 2, \ldots\}$ is an increasing filtration of $\sigma$-fields. For each $j \geq 1$, let $W_j$ be an $\mathcal{F}_j$-measurable Bernoulli random variable whose conditional success probability satisfies*

$$P(W_j = 1|\mathcal{F}_{j-1}) \geq \beta_j$$

*for some $\beta_j \in [0, 1]$. Then given $n \geq 1$,*

$$P\left(\sum_{j=1}^{n} W_j \leq (\sum_{j=1}^{n} \beta_j)/2\right) \leq \exp\left(-\frac{3\sum_{j=1}^{n} \beta_j}{28}\right) \tag{S6.2}$$

The proof for this can be found in Qian and Yang (2016).

**Lemma S8.** *Suppose $\{\mathcal{F}_j, j = 1, 2, \ldots\}$ is an increasing filtration of $\sigma$-fields. For each $j \geq 1$, let $\epsilon_j$ be an $\mathcal{F}_{j+1}$-measurable random variable that satisfies $E(\epsilon_j|\mathcal{F}_j) = 0$, and let $W_j$ be an $\mathcal{F}_j$-measurable random variable that is upper bounded by a constant $C > 0$ in absolute value almost surely. If there exists positive constants $v$ and $c$ such that for all $k \geq 2$ and $j \geq 1$, $E(|\epsilon_j|^k|\mathcal{F}_j) \leq k!v^2 c^{k-2}/2$, then for every $\epsilon > 0$ and every integer $n \geq 1$,*

$$P\left(\sum_{j=1}^{n} W_j \epsilon_j \geq n\epsilon\right) \leq \exp\left(-\frac{n\epsilon^2}{2C^2(v^2 + c\epsilon/C)}\right). \tag{S6.3}$$

233 *Proof of Lemma S8.* Lemma S8 is the same as Lemma 1 in Qian and Yang (2016) and the proof for
234 the same can be found there. □

235 A simplified version of Lemma S8 can be stated as follows.

236 **Corollary S3.** *Let $\epsilon_1, \epsilon_2, \ldots$ be independent random variables satisfying the refined Bernstein*
237 *condition, that is, if there exists positive constants $v$ and $c$ such that for all $k \geq 2$ and $j \geq 1$,*
238 *$E|\epsilon_j|^k \leq k!v^2 c^{k-2}/2$. Let $I_1, I_2, \ldots$ be Bernoulli random variables such that $I_j$ is independent of*
239 *$\{\epsilon_l : l \geq j\}$ for all $j \geq 1$. For any $\epsilon > 0$,*

$$P\left(\sum_{j=1}^n I_j \epsilon_j \geq n\epsilon\right) \leq \exp\left(-\frac{n\epsilon^2}{v^2 + c\epsilon}\right). \tag{S6.4}$$

240 The proof for this lemma can be found in Yang and Zhu (2002).

241 **Lemma S9.** *Suppose $\{\mathcal{F}_j, j = 1, 2, \ldots\}$ is an increasing filtration of $\sigma$-fields. For each $j \geq 1$,*
242 *let $\epsilon_j$ be an $\mathcal{F}_{j+1}$-measurable random variable that satisfies $E(\epsilon_j|\mathcal{F}_j) = 0$, and let $W_j$ be an $\mathcal{F}_j$-*
243 *measurable random variable that is upper bounded by a constant $C > 0$ in absolute value almost*
244 *surely. If $\epsilon_j \sim$ sub-Exp$(\nu^2, \alpha)$, then for every $\epsilon > 0$ and every integer $n \geq 1$,*

$$P\left(\sum_{j=1}^n W_j \epsilon_j \geq n\epsilon\right) \leq \begin{cases} \exp\left(-\frac{n\epsilon^2}{2C^2(v^2 + c\epsilon/C)}\right) & \text{, when } 0 < \epsilon < \frac{\nu^2 C}{\alpha} \\ \exp\left(-\frac{n\epsilon}{2\alpha C}\right) & \text{, when } \epsilon > \frac{\nu^2 C}{\alpha}. \end{cases} \tag{S6.5}$$