# OpenReview forum: "Regret Rates for $\epsilon$-Greedy Strategies for Nonparametric Bandits with Delayed Rewards"
_ICLR.cc/2024/Conference — ICLR 2024 Conference Withdrawn Submission_

### Official Review · Reviewer_S8Nn · 2023-10-30

**Soundness:** 3 good
**Presentation:** 1 poor
**Contribution:** 2 fair
**Rating:** 3
**Confidence:** 4

**Summary:**

In this paper we are dealing with a contextual bandit problem in which the reward are observed after a stochstic delay. The authors focus on the properties of the epsilon-greedy policy and provide regret bounds for such a strategy. They also provide some experimental results on the performance of two flavors of the algorithm and two adaptive strategies to automatically select from one strategy to the other.

**Strengths:**

The paper is formal and presents the theoretical results in a detailed manner.

I think that the paper might be of interest for the bandit community.

**Weaknesses:**

In the current status, I think that the paper is still not fluent enough to be considered for publication. First, the presentation could be improved by a lot. There are some suggestions below.

Moreover, the experimental part is not convincing, due to the lack of comparison with baselines (even non-delayed ones).

I think that in the introduction it would be more effective to cite only the most relevant works that relate to yours and defer the complete literature review in the following section. Regarding the literature review, it would help to have a clear definition of the problem (even informal) to understand the differences and similarities with what has been proposed in the literature.

Moreover, it is not clear why you decided to include the two lemmas in the main paper. Please add some comments or move them into the supplementary material.

If the derivation of the two results (Theorem 1 and 2) is significantly different, you should point out the elements of the proof that differ the most.

How does the theoretical result relate to the ones present in the literature (for similar settings)?

Regarding Figure, for no-regret algorithms, we should have that the curves are going to zero, but it would be hard to see this phenomenon. I would have divided the order of the bounds to show that the bounds are correct.

More comments on the results are required. Moreover, I would like to have a comparison with some baselines even the ones that are not dealing with delays.

**Questions:**

- Please provide more information about the application of your framework on the advertisement recommendation setting.
- Please refer to Algorithms in a proper way..."In Algorithm 1 we provide..."
- "are estimated nonparametrically..."This is the problem formulation, You should defer this comment to the methodology part
- "This could be done ..." do you have different theoretical results depending on the method you used? Please, comment on this aspect.
- "This choice is logical ..." I think that the explanation here is too simplistic. Please, provide more evidence fot your choice.
- Some of the assumptions are related to the problem, so they should be moved to the problem formulation. Others are related to the modeling approach you choose. These are not assumption but design choices for the algorithm you propose.
- It would be better to generalize Lemma2 and Lemma3 into a single lemma.
- You should also add some comment on the definition of n'_\delta
- it is somehow strange that you define a constant that is dependent on the time horizon N. I think you should remove the dependence from N to better highligh the dependence on the different parts of the regret from the time horizon N.
- It seems that setting the \pi_n sequence requires the modulus of continuity of the real function? Is it true? What if we do not have such an information on the true function?
- "Note that it is relevant ..." This comment should be moved to the introduction to better motivate the setting you are studying.
-

---

### Official Review · Reviewer_Zt6w · 2023-10-31

**Soundness:** 3 good
**Presentation:** 2 fair
**Contribution:** 2 fair
**Rating:** 5
**Confidence:** 3

**Summary:**

The paper discusses the critical issue of delayed feedback when implementing multi-armed bandit algorithms in real-world sequential decision-making scenarios. The authors delve deep into the realm of nonparametric contextual bandits that deal with delayed rewards. The paper introduces two different strategies for the $\epsilon$-greedy type allocation methods, each differing in how they adjust the exploration rate based on delays. By adopting the Nadaraya-Watson estimator for estimating mean reward functions, the strategies factor in unbounded random delays. The key contribution of the paper is the presentation of finite-time regret upper bounds for these strategies. In the backdrop of a comprehensive literature review on standard and contextual bandits with delayed rewards, the authors highlight the novelty of their work. Notably, this is the first instance where regret bounds for $\epsilon$-greedy in nonparametric bandits with delayed feedback are presented. From a practical standpoint, the authors propose data-driven methods to choose between the two strategies, ensuring that the most advantageous approach is selected based on the situation.

**Strengths:**

1. Theoretical Depth: The paper's rigorous introduction of two strategies, the associated finite-time regret upper bounds, and the employment of the Nadaraya-Watson estimator underscores its strong theoretical foundation.

2. Practical Considerations: Beyond the theoretical contributions, the authors present practical, data-driven strategies to adaptively select the best strategy, enhancing the usability of the proposed methods in real-world scenarios.

**Weaknesses:**

1. Dependence Structure: The adaptive strategies introduce an additional dependence structure, which could complicate its implementation. This poses new theoretical challenges that the authors recognize but leave unaddressed.

2. Limited Context: While the paper considers delays to be independent and unbounded, many practical situations could have delays dependent on the choice of arms or covariates. The strategies might not be directly applicable to such scenarios.

**Questions:**

1. How does the proposed $\epsilon$-greedy type allocation strategies compare in performance with other algorithms in the literature, especially in practical applications?

2. The paper introduces two strategies, $\eta_1$ and $\eta_2$. Could there be a hybrid approach that combines the best of both, rather than an adaptive selection between the two?

3. What is the computational overhead of the adaptive schemes when determining the most advantageous strategy between $\eta_1$ and $\eta_2$?

---

### Official Review · Reviewer_85Kj · 2023-11-01

**Soundness:** 3 good
**Presentation:** 3 good
**Contribution:** 2 fair
**Rating:** 5
**Confidence:** 3

**Summary:**

This paper studies $\epsilon$-greedy strategies for non-parametric bandits with delayed rewards. Compared to previous work [1], the main contribution of this paper is the derivation of finite-time regret bounds for the proposed algorithms. The authors validate their proposed algorithms through synthetic data experiments.

[1] Sakshi Arya and Yuhong Yang. To update or not to update? delayed nonparametric bandits with randomized allocation. Stat, 10(1):e366, 2021.

**Strengths:**

* The authors' theoretical analysis of the proposed algorithms appears to be sound. It is particularly interesting to note that the two proposed algorithms exhibit a tradeoff between estimation error and randomization error. Estimation error depends on the delay in receiving rewards, while randomization error is introduced to explore different actions. The authors propose a data-driven approach to switching between the two algorithms, which makes them well-suited for real-world applications.
* The authors evaluate the performance of their proposed algorithms through synthetic data experiments. The synthetic data is designed to mimic the characteristics of real-world data.

**Weaknesses:**

* The proposed $\epsilon$-greedy algorithms appear to be the same as those proposed in [1], which reduces the originality of the paper. If the authors' main contribution is the finite-time regret analysis, it might be better to emphasize the technical challenges of this analysis in the contribution paragraph.
* It would be beneficial if the authors could provide a lower bound analysis to further validate their contributions.

[1] Sakshi Arya and Yuhong Yang. To update or not to update? delayed nonparametric bandits with randomized allocation. Stat, 10(1):e366, 2021.

**Questions:**

Typos:

introduction: precision medicice -> precision medicine

Page 1: where the former consider... : consider -> considers

---

### Official Review · Reviewer_pTZn · 2023-11-01

**Soundness:** 4 excellent
**Presentation:** 3 good
**Contribution:** 3 good
**Rating:** 5
**Confidence:** 4

**Summary:**

This paper studies the problem of multi-armed bandits in a flexible non-parametric contextual bandits setting with delayed rewards.
To be specific, in this paper's setting, the rewards of each action is not revealed immediately, but only after some time, and in this paper's setting, the delay time is random and can be unbounded. The authors propose two epsilon-greedy-style algorithms with Nadaraya-Watson estimator for the above problem, and studies the regret upper bounds of the proposed algorithms. Due to the regret differences of these two algorithms, the authors also present an adaptive algorithm for find a balanced point between these two algorithms.

**Strengths:**

1. The setting in this paper is generic, fundamental and challenging. This setting can well cover a lot of hard and vague real-world problems. It tries to be as general as possible, despite some mathematical assumptions for the feasibility of the algorithms/regret bounds.

2. The writing and presenting are clear. The structure is logical. It is easy to follow.

**Weaknesses:**

It is not clear how good the regret results are without clear comparisons to previous works with similar settings or lower bounds. The epsilon-greedy-style algorithms also need compression or at least discussions with other styles of algorithms. Though due to the complexity of the setup, probably we do not have good understanding of the theoretical limits, but only from the existing results presented in this paper, it is hard for me to judge whether the results are good or not. The authors may strengthen the results by presenting the following:

1. A rough lower bound, which roughly matches the regret upper bound. The authors can state why the gap exist or why some terms are missing in the lower bound.

2. Comparison with the previous works with similar settings, such as with bounded/deterministic delays. Compare the upper bounds and state what terms are additional due to the randomness or unboundness.

3. A discussion about other classes of algorithms like UCB/Thompson sampling. Why these algorithms are not covered in this paper, and is there any challenge for them to be applied to this setting?

**Questions:**

Please refer to the suggestions in Section "Weakness"